# Habitat selection of female sharp-tailed grouse in grasslands managed for livestock production

**Megan C. Milligan**[1]*, **Lorelle I. Berkeley**[2], **Lance B. McNew**[1]

**1** Department of Animal and Range Sciences, Montana State University, Bozeman, Montana, United States of America, **2** Montana Department of Fish, Wildlife, and Parks, Helena, Montana, United States of America

* megan.milligan11@gmail.com

## Abstract

Habitat selection links individual behavior to population abundance and dynamics, so evaluation of habitat selection is necessary for conservation and management. Land management can potentially alter both the structure and composition of habitats, thus influencing habitat selection and population size. Livestock grazing is the dominant land use worldwide and, while overstocking has been linked to the decline of many wildlife species, properly managed grazing could improve habitat quality and maintain native rangeland habitats. We evaluated breeding season habitat selection of female sharp-tailed grouse, an indicator species for grassland ecosystems, in relation to grazing management and landscape features in eastern Montana and western North Dakota. At broad spatial scales, females selected for multiple landscape features, including grassland, but exhibited no selection for either landscape or management variables when selecting habitat at smaller spatial scales. Females selected for pastures managed with rest-rotation grazing when choosing a home range, but selection did not equate to improved fitness. Moreover, we observed strong individual variation in both home range size and third-order habitat selection. While the high variability among individuals makes specific management recommendations difficult, selection for grassland habitats at broad scales suggests that strategies that maintain intact native rangelands are important for the conservation of sharp-tailed grouse.

## Introduction

Habitat selection, especially for reproduction, is an important individual decision-making process that links individual behavior to population abundance and dynamics and determines the spatial distribution of both species and individuals [1–3]. The process of habitat selection represents a trade-off in which individuals balance competing demands, such as acquiring resources and avoiding predators, to maximize fitness [4, 5]. Thus, habitat selection is a key behavior that allows individuals to respond to spatial and temporal variation in their environment [6], and research increasingly suggests that both demography and habitat selection of wildlife populations vary spatially [7–10].

**Data Availability Statement:** All relevant data are within the paper and its Supporting Information files.

**Funding:** This research was supported by the general sale of hunting and fishing licenses in

Montana and matching funds under Federal Aid in Wildlife Restoration grant W-161-R1 awarded to LIB and LBM.

**Competing interests:** The authors have declared that no competing interests exist.

Land management has the potential to alter both the structure and composition of habitats and thus can influence the habitat selection of individuals and populations. Livestock grazing is the dominant land use worldwide and can influence the structure, composition, and productivity of habitats [11–14]. While overgrazing has been implicated in the decline of many wildlife species [13], properly managed grazing could benefit populations both by keeping native grasslands intact and providing a mosaic of habitats in different stages of disturbance, which may mimic historic disturbance regimes [11, 15, 16]. Specialized grazing systems that focus on enhancing structural and compositional heterogeneity in vegetation are being increasingly promoted and some systems, most notably patch-burn grazing, have been shown to benefit both wildlife and cattle production in tallgrass prairie ecosystems [16–21]. However, the effects of grazing on both wildlife and vegetation vary across broad spatial scales and are often strongly influenced by mediating factors such as precipitation and soil conditions [11, 22–25].

Rest-rotation grazing is a specialized grazing system that could function similarly to patch-burn grazing [26, 27] in areas like the northern Great Plains where fire is not a socially acceptable management technique [28], although this hypothesis has not been adequately tested. Originally developed to improve range condition [29], the rest-rotation system, developed by Hormay and Evanko [30], is based on the idea that grazing during consecutive growing seasons reduces plant vigor and that rest from grazing is necessary to allow plants to recover [29, 30]. By altering the timing of grazing for individual pastures each year and incorporating an additional period of rest, rest-rotation grazing could also create a patchwork of habitats on the landscape, with rested pastures having the most residual cover [26, 27]. By utilizing a patchwork of habitats, individuals may be able to better balance foraging activities with predator avoidance. The effects of grazing system, however, are also influenced by stocking rate, which is a measure of grazing intensity, and high stocking rates can have negative effects, particularly on grouse [17, 18, 25, 31].

Recognized as an indicator species for grassland ecosystems [32], sharp-tailed grouse (*Tympanuchus phasianellus*) are a model species to evaluate the effects of livestock grazing on wildlife. Throughout their life history, sharp-tailed grouse have diverse habitat requirements, including short, bare areas for lekking, denser herbaceous cover for nesting, and deciduous shrubs for winter cover and food. Identifying management strategies to conserve grouse populations could improve conservation of a variety of other grassland species [33, 34]. However, very little is known about the general spatial ecology of sharp-tailed grouse and specifically the effects of livestock grazing on their space use.

Habitat selection by prairie-grouse (*Tympanuchus* spp.) is driven in part by predator avoidance, so having sufficient cover is important to conceal both nests and adults [35]. Therefore, grazing, which can influence both the structure and composition of habitats, could have important indirect effects on grouse selection behavior. Patch-burn grazing, a management strategy that increases heterogeneity in tallgrass prairies, improved habitat for greater prairie-chickens (*T. cupido*) and lesser prairie chickens (*T. pallidinctus*) relative to management that incorporates annual spring burning and intensive early stocking [36, 37]. Beyond the effects of patch-burn grazing on prairie-chickens, however, the effects of livestock grazing on prairie-grouse are not well understood [38].

Other factors, such as landscape configuration and anthropogenic development, can also influence habitat selection. Grouse have been shown to minimize predation risk at multiple spatial scales by selecting for habitats providing horizontal and vertical cover [39–41], sites with more grassland on the landscape [42–44], and less cropland [45, 46, but see 39]. Other studies, however, suggest that landcover does not have a large influence on selection or that selection for different habitat types varies among sites [43, 46, 47]. Anthropogenic development generally has negative effects on grouse. Greater sage-grouse (*Centrocercus*

*urophasianus*) selected for lower densities of oil and gas development, sharp-tailed grouse avoided roads and distribution lines [48], and greater and lesser prairie-chickens avoided anthropogenic structures and expanded home ranges in proximity to wind energy development [47, 49, 50]. Home range size was not related to road density, however, and selection for or against roads varied among study areas for prairie-chickens [43]. Further complicating relationships, aspects of habitat selection can change from year to year with different weather conditions [36], and can vary across spatial scales, with home range size for prairie-chickens, for example, related to the amount of precipitation received at different sites spread across multiple states [43]. Furthermore, habitat selection can vary with the availability of a resource, termed a functional response, where individuals experiencing different conditions may respond differently [51]. Taken together, the lack of information for sharp-tailed grouse and the differing results for related species across time and space make generalized habitat management recommendations inappropriate.

Our objective was to evaluate the effects of livestock grazing management on the breeding season habitat selection of female sharp-tailed grouse while considering other habitat features at multiple orders of selection. Habitat selection is a hierarchical process and studies that evaluate selection at multiple spatial scales can improve understanding of wildlife-habitat relationships [52, 53]. We evaluated both second- and third-order habitat selection of female grouse, defined as the selection of habitat for an individual's home range within the larger study area and the selection of habitat within an individual's home range, respectively [52]. Livestock grazing has the potential to maintain grassland habitats [54] and we hypothesized that grouse would select for large grassland patches at all orders of selection. Furthermore, rest-rotation grazing could influence grouse habitat selection by creating a patchwork of habitats that are periodically rested from disturbance. Therefore, we hypothesized that if rest-rotation grazing increases heterogeneity in grassland habitats, then females would select for rest-rotation pastures and have smaller home ranges when using those potentially higher-quality pastures due to increased availability or proximity of important resources.

## Study area

This study was conducted during 2016–2018 in southern Richland and McKenzie Counties in eastern Montana and western North Dakota, U.S.A., respectively (centered on 47.52˚N, -104.06˚W). The study area was composed of Great Plains mixed-grass prairie interspersed with Great Plains badlands and wooded draws and ravines [55] and was primarily managed for cattle production. Vegetation was a mixture of mid and short grasses, with western wheatgrass (*Pascopyrum smithii*), little bluestem (*Schizachyrium scoparium*), needle-and-thread (*Hesperostipa comata*), Kentucky bluegrass (*Poa pratensis*), blue grama (*Bouteloua gracilis*), and crested wheatgrass (*Agropyron cristatum*) being the dominant graminoids. The three study years differed drastically in the amount of precipitation received. We obtained daily precipitation data from the National Oceanic and Atmospheric Association (NOAA) station in Sidney, MT, and calculated the amount of precipitation received annually (1 January–31 December) and during the sharp-tailed grouse breeding season (15 March–15 August). Annual precipitation was 419.3 mm in 2016, 216.4 mm in 2017, and 341.5 mm in 2018. Total precipitation during the breeding season was 268.7 mm in 2016, 105.2 mm in 2017, and 312.1 mm in 2018.

The study area was centered on an ~3,300-ha Upland Gamebird Enhancement Program (UGBEP) project established by the Montana Department of Fish, Wildlife and Parks in 1993 that included four separate 3-pasture Hormay rest-rotation systems (Hormay and Evanko 1958). In a given year, cattle were stocked in one pasture from May—July (growing season),

then moved to a second pasture during August—October (post-growing season), while the third pasture was rested and the order of rotation was shifted within each 3-pasture rest-rotation system the next year. Therefore, no pasture was grazed during the same season in consecutive years and pastures rested in the previous year theoretically should have had the most residual cover. Average pasture size in the four rest-rotation systems was 292 ± 116 ha. Pastures of surrounding ranches, which included both private land and 4 pastures located on U.S. Forest Service National Grasslands were managed with more commonly used livestock grazing systems, including both season-long systems (19 pastures, ~4,800 ha) and 2- and 3-pasture summer rotation systems (25 pastures, ~5,200 ha). Grazing occurred in season-long pastures from approximately May to early November, while cattle were stocked in the same pastures in summer rotation systems each year for the same 6–8-week period (approximately April–June, June–July or Aug–Nov). Average pasture sizes in the season-long and summer rotation systems were 242 ± 312 ha and 238 ± 335 ha, respectively. Stocking rates were controlled by landowners and lessees and averaged rates were 0.93 animal unit month (AUM) ha$^{-1}$, 1.46 AUM ha$^{-1}$, and 0.76 AUM ha$^{-1}$ for rest-rotation, season-long, and summer rotation pastures, respectively. The range of stocking rates for grazed pastures was 0.38–3.25 AUM ha$^{-1}$, 0.17–4.28 AUM ha$^{-1}$, and 0.21–4.45 AUM ha$^{-1}$ for rest-rotation, season-long, and summer rotation pastures, respectively, and included similar distributions within each grazing system [56]. Average stocking rates did not exceed the range of rates (1.11–1.48 AUM ha$^{-1}$) recommended by the Natural Resources Conservation Service (NRCS) for the most common ecological site (R058AE001MT) in the study area. Environmental variables including topography, average vegetation productivity, soil type, vegetation canopy greenness as measured by the Normalized Difference Vegetation Index (NDVI) in June 2018, and the variation in small-scale vegetation cover and structure were similar among grazing systems [56].

## Methods

We captured grouse using walk-in funnel traps at 12 leks (5 in rest-rotation pastures, 3 in summer rotation pastures, and 4 in season-long pastures) during March—May in 2016–2018. Females were fitted with very high frequency (VHF) radio-transmitters (model A4050; Advanced Telemetry Systems, Isanti, MN). Radio-marked females were located by triangulation or homing ≥ 3 times/week during the breeding season (15 March– 15 August). Coordinates for triangulated locations were calculated using Location of a Signal software (LOAS; Ecological Software Solutions LLC, Hegymagas, Hungary) and examined for spatial error. All locations with low estimation precision (> 200 m error ellipse) were discarded. All animal handling was approved under Montana State University's Institutional Animal Care and Use Committee (Protocol #2016–01) and permits for field studies were obtained from both Montana Fish, Wildlife and Parks and North Dakota Game and Fish.

We analyzed location data for the breeding season (15 March– 15 August) and defined a home range as the space an individual needed to forage, reproduce, and survive. Previous studies have found that small sample sizes can bias home range estimates [57], so analyses were restricted to birds with ≥ 30 locations and ≥ 20 locations not associated with a nest site. We used the fixed kernel method [58] with the default smoothing parameter to calculate home ranges as the 95% utilization distribution for the breeding season (April—August) using the adehabitatHR package in Program R v3.5.1.

We identified nine landscape metrics *a priori* that could influence sharp-tailed grouse space use. Three of those metrics were related to rangeland management: grazing system and stocking rate (AUM ha$^{-1}$) during either the current or previous year. Two landscape metrics represented anthropogenic disturbance, including oil pads and roads. Four additional landscape

variables were related to landcover: % grassland, % wooded draws, % cropland, and the density of edge habitat (total landcover edge length / polygon area). We collected information on grazing management for every pasture in the study area by interviewing landowners to determine the number and class of animals stocked and the timing of stocking to determine the grazing system (rest-rotation, summer rotation, season-long) and stocking rate (AUM ha$^{-1}$) during the current and previous year. Stocking rate is a measure of the number of animals in a unit area (e.g., pasture) during the entire grazing season. We digitized the location of oil pads and roads in the study area and roads were defined as paved and dirt state and county roads and did not include ranch two-tracks. We utilized the 30-m resolution LANDFIRE data depicting landcover type for habitat classifications [55]. A habitat patch edge was defined as any area where the landcover type (grassland, wooded draws, or cropland) of adjacent pixels was different and edge density was defined as the length of patch edge divided by home range size.

We examined second order selection, or an individual's selection of a home range within the larger study area, using resource selection functions to compare used and available home ranges following Design II of Manly et al. [59]. We characterized grouse resource use with estimated home ranges for each individual for each year. If an individual was monitored in multiple years, we randomly selected one home range to include in analyses. To sample availability, we randomly placed 1,000 circular home ranges across the study area that were equal in area to the average grouse home range size (~500 ha). The study area was defined as the 99% kernel utilization distribution estimated using locations from all collared individuals. Using the spatial layers described above, we calculated the following variables within each used and available home range: proportion grassland, proportion wooded draws, proportion cropland, total edge density, average distance to oil pad, average distance to road, and the proportion of each grazing system. We then examined correlations for each pair of explanatory variables ($r > |0.6|$), and excluded proportion cropland and edge density, which were both highly collinear with proportion grassland. We then used logistic regression to compare used and available home ranges with available home ranges weighted ($w = 1000$) to improve convergence [60]. We first evaluated all combinations of habitat and anthropogenic variables, and then built a final candidate model set including a top habitat and anthropogenic model in combination with all combinations of grazing management variables. We compared models based on Akaike's Information Criterion for small sample sizes (AIC$_c$) and models representing the majority of model weight ($w_i$) were considered the most important [61]. We considered variables to be significant if 85% confidence intervals did not overlap zero and variables were considered uninformative if a model was <2 ΔAIC$_c$ units from the top model but only differed in a single parameter [62].

We used linear models to evaluate the relationship between home range size and each metric described above, as well as the effects of year; nest outcome; and distance to nearest lek. We evaluated all single-variable models using Akaike's Information Criterion corrected for small sample sizes (AIC$_c$). Models that were within 2 ΔAIC$_c$ of the top model and represented a majority of model weight ($w_i > 0.6$) were considered important.

To evaluate third-order habitat selection, or the selection of habitat within individual home ranges, we used resource selection functions to compare used and available points following Design III of Manly et al. [59], where individual telemetry locations were classified as used points and available points were randomly sampled for each individual within their home range. We evaluated the nine landscape metrics described above at both used and available points. For the habitat variables, we used FRAGSTATS 4.2 [63] to conduct a moving window analysis to calculate the proportion of each landcover type and the density of edge habitat within 8 buffer distances representing various spatial scales of influence (30, 75, 125, 200, 500, 750, 1000, 1300 m) and evaluated the scale for each landcover type that best predicted grouse

space use [64]. We chose scales across a continuum, with 30 m representing the minimum size imposed by our spatial data and 1,300 m approximating the average size of the breeding season home range of a female sharp-tailed grouse in our study area. A scale of 200 m represents the average distance moved daily by female sharp-tailed grouse during the breeding season in our study. The remaining scales represent intermediate distances between the minimum imposed by our spatial data and the average size of a breeding season home range.

We conducted 1,000 simulations for each variable and each scale of landcover variables to determine the number of available points required for coefficient estimates to converge [S1–S5 Figs; 60]. Based on the simulations, available points were sampled at a 15:1 available:used ratio within each individual bird's home range to balance coefficient convergence and computational efficiency. For all models, we used binomial linear mixed models in a Bayesian framework with both random intercepts and slopes to account for potential autocorrelation among sampling points and individual variation in selection [65, 66]. For the four landcover covariates, we first selected the spatial scale at which selection was the strongest. We compared the 8 buffer distances using calculated deviance information criteria (DIC) to identify a top model *sensu* Laforge et al. [64], and we considered > 5 DIC units to be a substantial difference in model fit [66].

After assessing collinearity for each pair of explanatory variables ($r \geq 0.6$) and selecting the variable with the most support based on calculated DIC, we then evaluated support for all management and landscape variables in a full model. We centered and scaled all predictor variables to calculate standardized coefficients of fixed effects to make population-level inferences about each habitat variable and improve model convergence. Coefficients with 95% credible intervals that did not overlap zero were considered important. To determine the degree of variation in selection among individuals, we examined variation in individual-specific slopes for each predictor variable and calculated the number of individuals that were significantly selecting for or against each variable based on 95% credible intervals. To evaluate whether selection varied with resource availability, we calculated the mean value of each covariate at used and available points for each individual female [67, 68] and plotted the use of a variable against its availability [51].

We fit all binomial selection models with random intercepts and slopes using integrated nested Laplace approximation (INLA) using the R-INLA package in Program R. This approach is a computationally efficient alternative to existing algorithms because it circumvents Markov chain Monte Carlo (MCMC) sampling by providing efficient approximations of marginal posterior distributions and it has been shown to be useful for fitting generalized linear mixed models used to calibrate resource selection functions [69–71]. Following recommendations from Muff et al. [69], we used independent priors with large prior variance for all model components and used penalized complexity priors for the variances of random slopes (see example code in S1 Appendix). Because individual-specific intercepts are not of interest in resource selection functions, we treated them as random effects with large, fixed variance ($10^6$) following Muff et al. [69].

## Results

During the 2016–2018 breeding seasons, we collected a total of 7,178 locations and calculated 142 home ranges for 118 individual females (40 in 2016, 53 in 2017, 49 in 2018). Home range size was estimated without bias relative to sampling effort (S1 Table). Mean breeding season home range size for all females was 489 ± 41 ha but varied from 58–3,717 ha (Table 1). Home range sizes were less variable within pastures managed with summer rotation grazing compared to those in other systems (Fig 1), but grazing system did not have a significant effect on

**Table 1. Home range size (95% volume contour) for radio-marked female sharp-tailed grouse monitored in the 3 grazing systems during the breeding seasons of 2016–2018.** Females were assigned to the grazing system containing $\geq$ 60% of their home range or were considered to use multiple systems if no one system accounted for $\geq$ 60% of their home range.

| Grazing System | # Females | Mean area (ha) ± SE | Min. area (ha) | Max area (ha) |
|---|---|---|---|---|
| Rest-rotation | 47 | 557 ± 94 | 63.81 | 3717.45 |
| Summer rotation | 44 | 361 ± 39 | 86.13 | 1198.89 |
| Season-long | 36 | 408 ± 43 | 57.51 | 1103.58 |
| Multiple systems | 15 | 838 ± 179 | 191.43 | 2265.66 |

average size of home ranges (Table 2). Density of edge habitat within the home range was the best predictor of home range size (Table 2) and was negatively related to the size of breeding season home ranges ($\beta$ = -5.26 ± 1.48; Fig 2).

At the second order, breeding season home range selection was best predicted by the proportion grassland ($\beta$ = 0.48 ± 0.13), the proportion wooded draws ($\beta$ = 0.30 ± 0.11), distance to oil pad ($\beta$ = 0.32 ± 0.11), and the proportion rest-rotation ($\beta$ = 0.24 ± 0.10) within the home range (Table 3 and S2 Table). The proportion grassland had the strongest effect based on scaled coefficients, but all variables had positive effects on home range selection. The relative

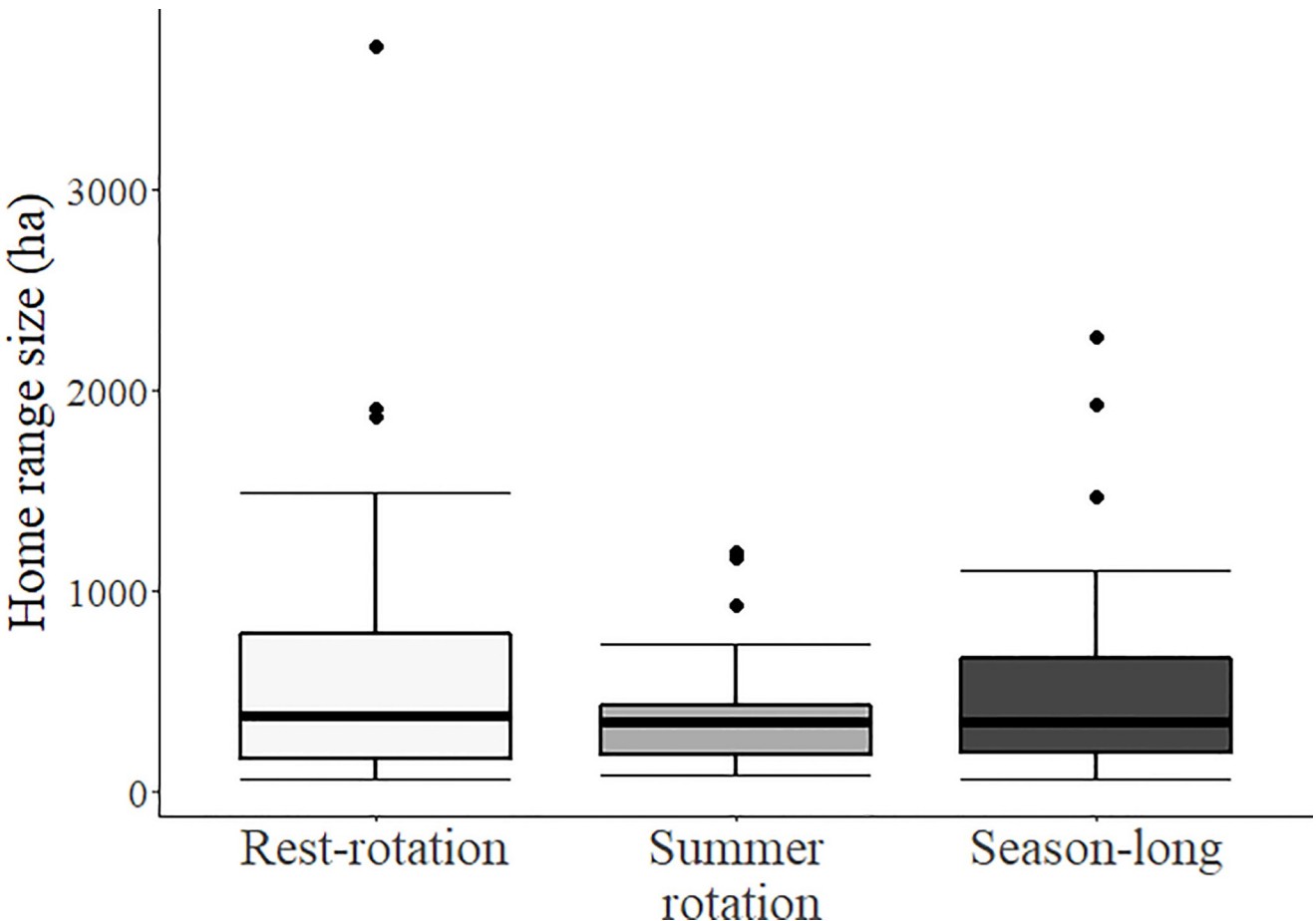

**Fig 1. Female sharp-tailed grouse breeding season home range size (± SE) by grazing system.** An individual female was assigned to a grazing system according to the system containing $\geq$ 60% of the individual's home range.

**Table 2. Support for candidate models predicting the home range size of female sharp-tailed grouse during the breeding seasons of 2016–2018.** The percent of a home range containing either the rest-rotation or summer rotation system are measured in relation to the season-long system. The number of parameters (K), $AIC_c$ values, $\Delta AIC_c$ values, model weights ($w_i$), and log-likelihoods are reported.

| Model | K | $AIC_c$ | $\Delta AIC_c$ | $AIC_c$ $w_i$ | LogLik |
|---|---|---|---|---|---|
| Edge density | 3 | 2157.27 | 0.00 | 0.93 | -1075.55 |
| Dist. to grassland edge | 3 | 2165.05 | 7.78 | 0.02 | -1079.44 |
| Nest outcome | 3 | 2165.25 | 7.98 | 0.02 | -1079.54 |
| Null | 2 | 2166.80 | 9.53 | 0.01 | -1081.36 |
| Year | 3 | 2167.47 | 10.20 | 0.01 | -1080.65 |
| % Rest-rotation | 3 | 2167.71 | 10.43 | 0.01 | -1080.77 |
| Stocking rate | 3 | 2168.12 | 10.84 | 0.00 | -1080.97 |
| % Summer rotation | 3 | 2168.14 | 10.87 | 0.00 | -1080.98 |
| Dist. to lek | 3 | 2168.65 | 11.38 | 0.00 | -1081.24 |
| Dist. to road | 3 | 2168.73 | 11.46 | 0.00 | -1081.28 |
| Dist. to oil pad | 3 | 2168.84 | 11.57 | 0.00 | -1081.33 |
| Prop. grassland | 3 | 2168.88 | 11.61 | 0.00 | -1081.36 |

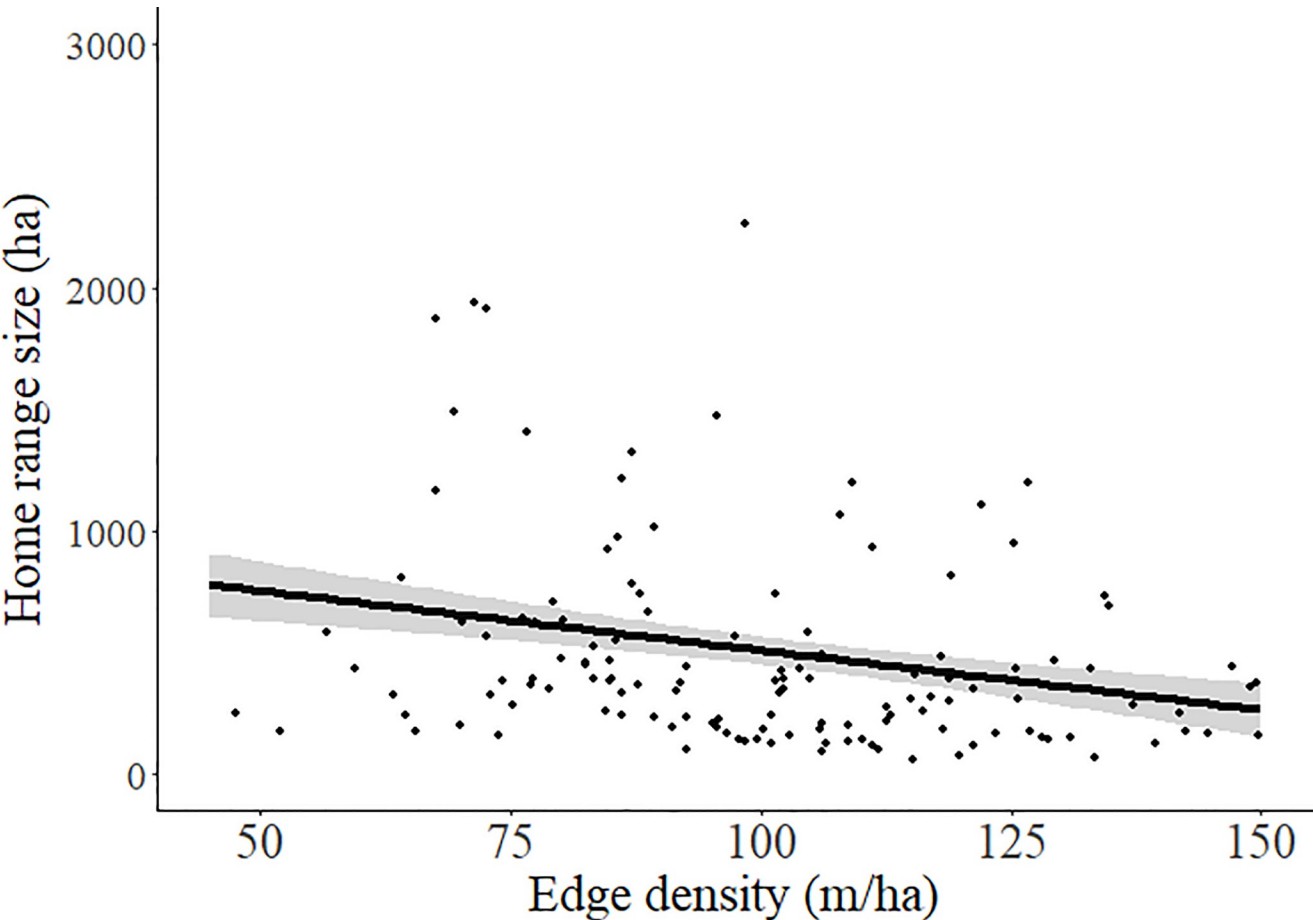

**Fig 2. Relationship (± 85% confidence intervals) between the density of edge habitat (total landcover edge length / polygon area) and breeding season home range size for female sharp-tailed grouse.**

**Table 3. Support for candidate models predicting second order selection, or home range selection, of female sharp-tailed grouse during the breeding seasons of 2016–2018.** The percent of a home range containing either the rest-rotation or summer rotation system are measured in relation to the season-long system. The number of parameters (K), $AIC_c$ values, $\Delta AIC_c$ values, model weights ($w_i$), and log-likelihoods are reported.

| Model | K | $AIC_c$ | $\Delta AIC_c$ | $AIC_c$ $w_i$ | LL |
|---|---|---|---|---|---|
| % Grassland + % wooded draws + dist. to oil pad + % rest-rotation | 5 | 2322.10 | 0.00 | 0.44 | -1156.02 |
| % Grassland + % wooded draws + dist. to oil pad + % rest-rotation + current stocking rate | 6 | 2323.92 | 1.83 | 0.18 | -1155.92 |
| % Grassland + % wooded draws + dist. to oil pad + % rest-rotation + previous stocking rate | 6 | 2324.08 | 1.98 | 0.17 | -1156.00 |
| % Grassland + % wooded draws + dist. to oil pad | 4 | 2326.07 | 3.98 | 0.06 | -1159.02 |
| % Grassland + % wooded draws + dist. to oil pad + % summer rotation | 5 | 2326.27 | 4.17 | 0.06 | -1158.11 |
| % Grassland + % wooded draws + dist. to oil pad + current stocking rate | 5 | 2327.64 | 5.54 | 0.03 | -1158.79 |
| % Grassland + % wooded draws + dist. to oil pad + previous stocking rate | 5 | 2328.07 | 5.97 | 0.02 | -1159.01 |
| % Grassland + % wooded draws + dist. to oil pad + % summer rotation + current stocking rate | 6 | 2328.07 | 5.97 | 0.02 | -1158.00 |
| % Grassland + % wooded draws + dist. to oil pad + % summer rotation + previous stocking rate | 6 | 2328.28 | 6.19 | 0.02 | -1158.10 |
| % Rest-rotation | 2 | 2334.22 | 12.12 | 0.00 | -1165.10 |
| % Rest-rotation + previous stocking rate | 3 | 2335.96 | 13.87 | 0.00 | -1164.97 |
| % Rest-rotation + current stocking rate | 3 | 2335.99 | 13.89 | 0.00 | -1164.98 |
| Null | 1 | 2336.38 | 14.29 | 0.00 | -1167.19 |
| % Summer rotation | 2 | 2338.20 | 16.10 | 0.00 | -1167.09 |
| Current stocking rate | 2 | 2338.38 | 16.28 | 0.00 | -1167.18 |
| Previous stocking rate | 2 | 2338.39 | 16.29 | 0.00 | -1167.19 |
| % Summer rotation + current stocking rate | 3 | 2340.17 | 18.07 | 0.00 | -1167.07 |
| % Summer rotation + previous stocking rate | 3 | 2340.20 | 18.10 | 0.00 | -1167.09 |

probability of home range selection increased with more grassland, more wooded draws, further from oil pads, and in pastures managed with rest-rotation grazing (Fig 3).

At the third order, preliminary analyses suggested that a spatial scale of 1,300 m for grassland, 1,300 m for wooded draws, 500 m for cropland, and 1,000 m for edge density represented the scale of strongest female habitat selection (S3 Table). However, the proportion of grassland was correlated with both the proportion of cropland and the density of edge habitat (S4 Table), so only proportion grassland was used in the full model. In the full analysis, 95% credible intervals for all variables overlapped zero, suggesting no significant selection at the population level (Fig 4). However, variability in selection as measured by the variation in individual-specific slopes for each predictor variable was high, indicating large differences in individual habitat selection (Fig 5). Individuals were selecting for and against habitat variables in equal numbers (Fig 6), resulting in no population-level selection. Nevertheless, selection varied linearly with availability suggesting that habitat use was proportional to availability (Fig 6).

Symbols represent individual females that selected against (gray circle), selected for (gray triangle), or displayed no significant selection (white square) for each variable and the diagonal represents proportional resource use.

## Discussion

High individual variability in both home range size and third-order habitat selection of female sharp-tailed grouse outweighed any potential population-level trends. When selecting home ranges, females strongly selected for multiple landscape features, whereas third-order selection within home ranges was highly variable among individuals but proportional to availability, which suggests highly plastic habitat use within the population at this scale. While grouse selected for pastures managed with rest-rotation grazing when selecting a home range, we

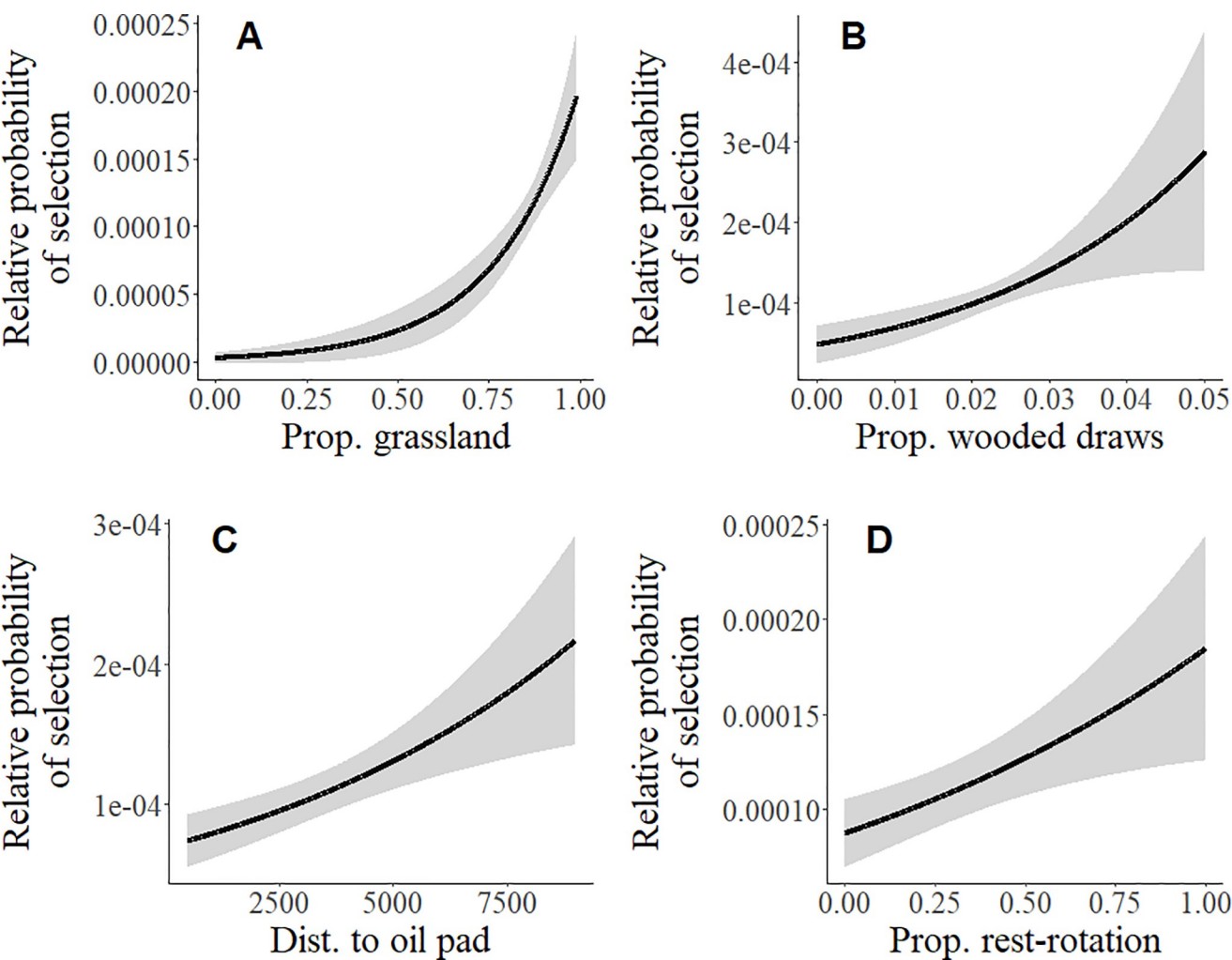

**Fig 3. Relationship (± 85% confidence intervals) between the proportion grassland (A), the proportion wooded draws (B), distance to oil pad (C), and the proportion rest-rotation (D) and the relative probability of breeding season home range selection for female sharp-tailed grouse.**

found no evidence for selection based on grazing management when choosing locations within home ranges.

Home range sizes in our study were on average larger and more variable than those previously reported for sharp-tailed grouse, although previous studies were limited by sample size and often included male grouse [41, 72, 73]. Previous estimates of home ranges for sharp-tailed grouse have come primarily from shrub-steppe or forested regions and our home range estimates are more in line with those from prairie-chickens in the Great Plains that had larger home ranges with more variation among individuals [36, 43, 74]. Home range size was negatively related to the density of edge habitat, suggesting that females could use a smaller area to meet their basic needs in more diverse habitats. At this scale, females strongly selected for grassland, which is consistent with previous studies finding both general selection for grassland [42–44] and that increased cropland on the landscape decreased adult survival in our study area [75], although the negative relationship between home range size and edge density and selection for wooded draws may suggest that other habitat types are important to female

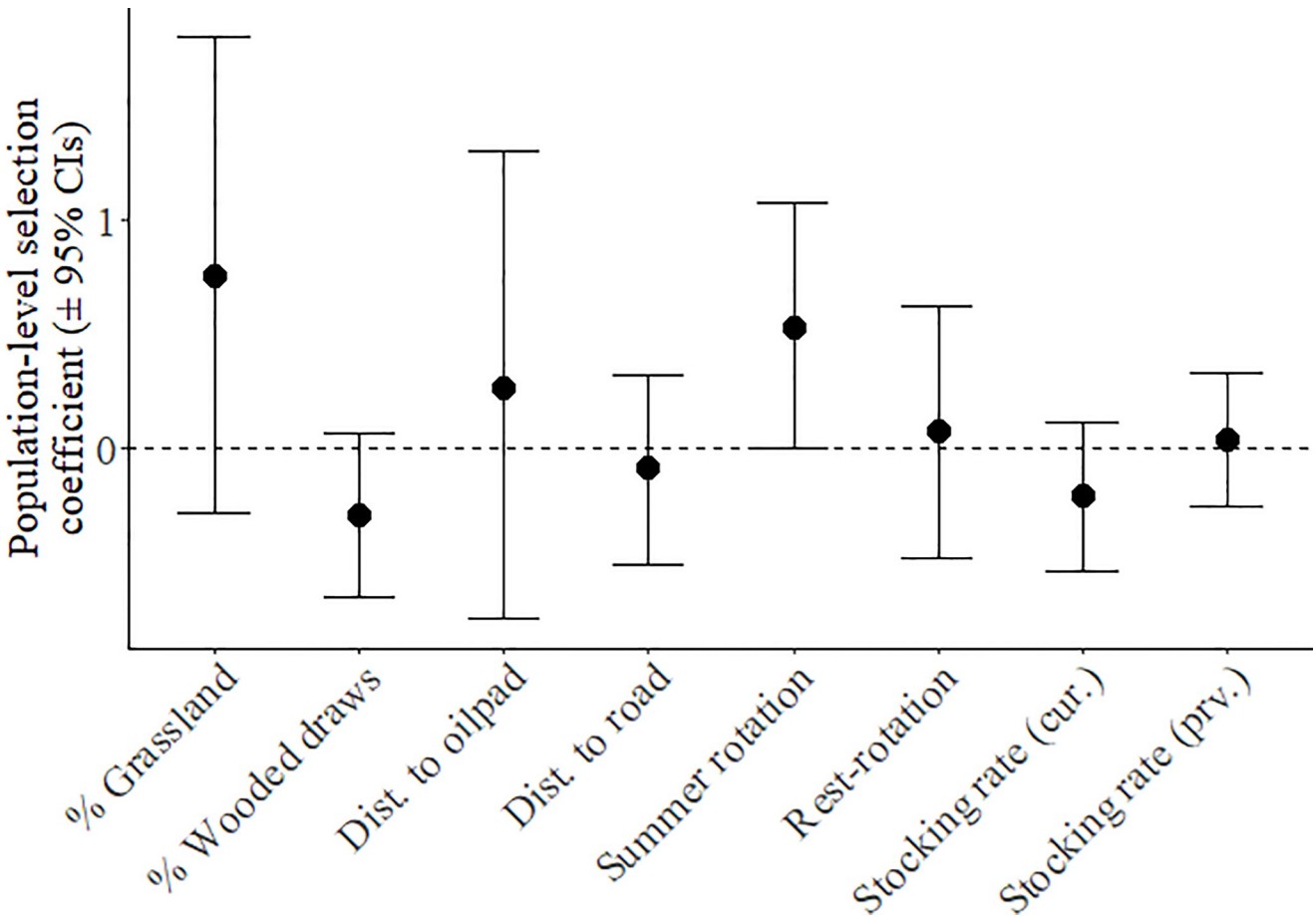

**Fig 4. Fixed effects representing population-level habitat selection of sharp-tailed grouse within their home ranges during the breeding season, with error bars representing 95% credible intervals.**

grouse during the breeding season. In addition, females selected home ranges that were further from oil pads, which supports previous research that has found consistently negative effects of energy development on grouse [47, 49, 50, 76].

When choosing a home range, most females selected for pastures managed with rest-rotation grazing but showed no selection for either grazing system or stocking rate when selecting habitat within the home range. Our results corroborate previous research finding that greater prairie-chickens strongly selected for areas managed with a heterogeneity-based fire-grazing management system [36]. However, previous research in our study area found that grazing system was not strongly linked to nest survival or adult female survival, important demographic parameters influencing grouse populations [56, 77]. Thus, selection for rest-rotation pastures did not equate to improved fitness for nesting sharp-tailed grouse. We found no evidence that space use of sharp-tailed grouse was influenced by stocking rate, which conflicts with previous studies that have documented consistently negative effects of high stocking rates on prairie-chickens [17, 18, 31]. Stocking rates in our study area were considered light to moderate by NRCS standards though [78] and it is possible that selection may only be apparent across a broader range of stocking rates.

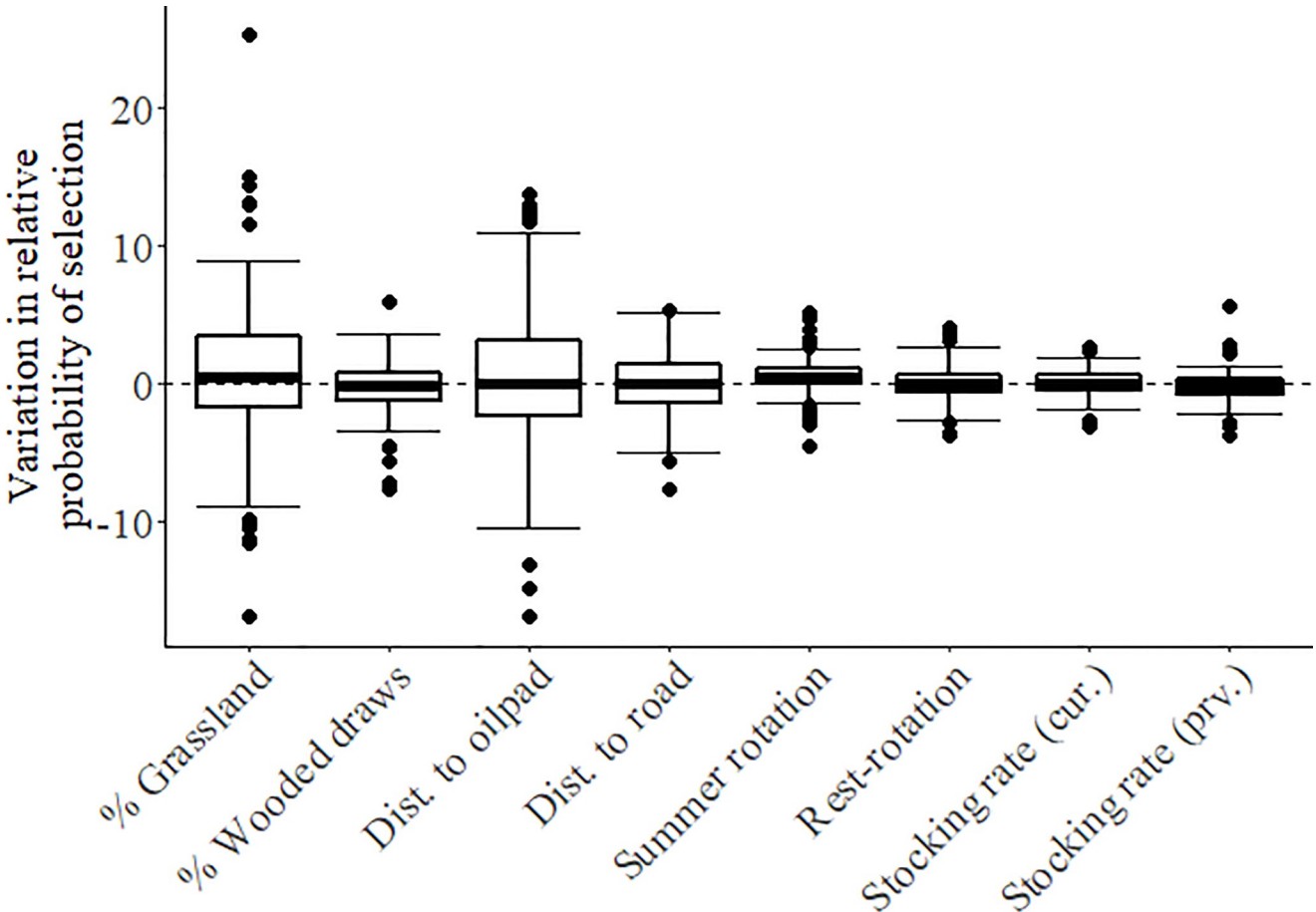

**Fig 5. Variation in individual-specific slopes for each variable evaluated in the third order habitat selection analysis.**

In contrast to selection of home ranges, we found no evidence of selection for habitat features within the home range and our results conflict with previous studies observing small-scale selection based on vegetation features [39, 41, 79]. Our habitat variables consisted only of remotely-sensed data, however, and did not include fine-scale measures of vegetation structure or composition; a related study found that small-scale vegetation was critical to nest survival and selection [56]. Furthermore, grouse habitat selection based on both landcover and anthropogenic disturbance such as roads has been shown to vary among studies and even sites within a single study [36, 39, 42, 43, 46, 47], which can complicate population-level interpretation of effects.

While there was no evidence for population-level selection at the third order, there was significant individual variation in habitat selection within the home range, suggesting that fine-scale habitat selection may be flexible or less important after a home range has been selected. Significant individual variation is consistent with previous work suggesting that habitat selection can vary by year or weather conditions and can vary across spatial scales [36, 43]. Taken together, this suggests that generalized habitat recommendations across sites and related species may not be appropriate. Given that habitat use did not vary with availability, the variation in habitat selection behavior suggests a high degree of plasticity in the population [80]. If individual differences are consistent across time, then those differences can represent alternative

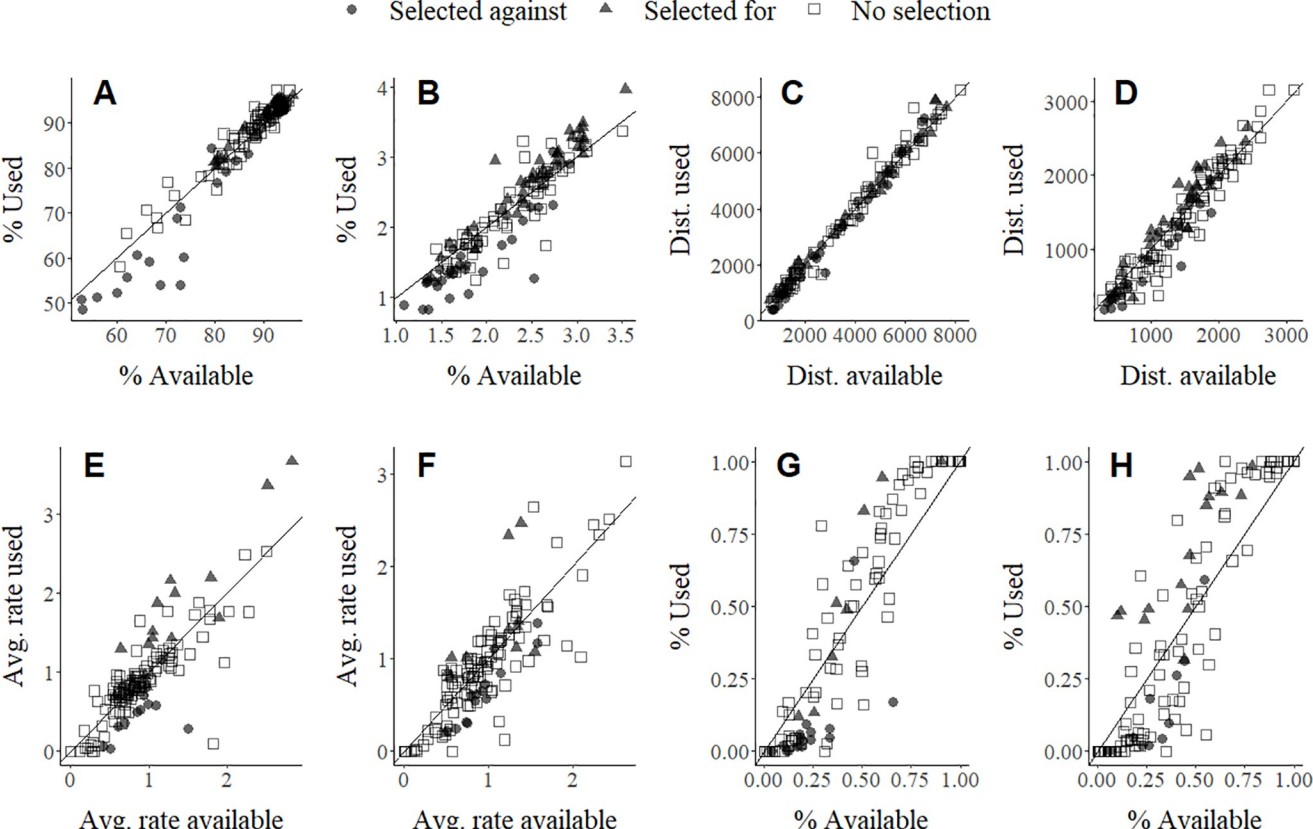

**Fig 6. Mean habitat values at used relative to available points for % grassland (A), % wooded draws (B), distance to oil pad (C), distance to road (D), current stocking rate (E), previous stocking rate (F), % summer rotation (G), and % rest-rotation (H) for individual female sharp-tailed grouse selecting habitat within home ranges.**

approaches that evolved to respond to a variable environment [81, 82]. Regardless, if individual differences are correlated with fitness, individual variation can have ecological and evolutionary implications [83, 84]. Future research should explore both the consistency in individual differences in resource selection across time and the link between individual differences and fitness.

## Conclusions

At a broad scale, female sharp-tailed grouse exhibited strong selection, particularly for grassland, when choosing a home range, but showed no selection for habitat or management variables when selecting locations within their home ranges. Females did select home ranges in pastures managed with rest-rotation grazing, but selection was not related to improved reproductive success or survival [56, 77]. Given observed individual variation, the choice of grazing system may not have a significant influence on sharp-tailed grouse populations in the northern mixed-grass prairie when stocking rates are low to moderate. Importantly, female sharp-tailed grouse exhibited strong individual differences in both home range size and third-order habitat selection that outweighed any potential population-level trends, suggesting that specific management recommendations are inappropriate, particularly across large spatial scales. Collectively, our results suggest that maintaining large intact grasslands on the landscape will have higher conservation value for sharp-tailed grouse than prescriptive livestock grazing systems.

## Supporting information

**S1 Appendix. Example code for Bayesian logistic regression model evaluating third order habitat selection using the R package R-INLA.**
(DOCX)

**S1 Fig. Simulation results evaluating the number of available points necessary for convergence of the proportion of grassland measured at different buffer distances.**
(TIF)

**S2 Fig. Simulation results evaluating the number of available points necessary for convergence of the proportion of wooded draws measured at different buffer distances.**
(TIF)

**S3 Fig. Simulation results evaluating the number of available points necessary for convergence of the proportion of row crop agriculture measured at different buffer distances.**
(TIF)

**S4 Fig. Simulation results evaluating the number of available points necessary for convergence of edge density measured at different buffer distances.**
(TIF)

**S5 Fig. Simulation results evaluating the number of available points necessary for convergence of variables measured at a single scale.**
(TIF)

**S1 Table. Support for candidate models predicting the relationship between the number of locations per female and home range size of female sharp-tailed grouse during the breeding seasons of 2016–2018.** The number of parameters (K), $AIC_c$ values, $AIC_c$ values, model weights ($w_i$), and log-likelihoods are reported.
(DOCX)

**S2 Table. Support for candidate models predicting the relationship between habitat and anthropogenic variables and home range selection of female sharp-tailed grouse during the breeding seasons of 2016–2018.** The number of parameters (K), $AIC_c$ values, $AIC_c$ values, model weights ($w_i$), and log-likelihoods are reported.
(DOCX)

**S3 Table. Support for models predicting the spatial grain of each landcover variable that best predicts sharp-tailed grouse habitat selection during the breeding seasons of 2016–2018, based on Deviance Information Criteria (DIC).**
(DOCX)

**S4 Table. Multicollinearity results for management and landscape variables in the full third order resource selection analysis evaluating habitat selection within the home range for sharp-tailed grouse during the breeding seasons of 2016–2018.**
(DOCX)

**S1 Data.**
(ZIP)

## Acknowledgments

Our study was made possible by the generous cooperation of private landowners who allowed us access to their land and the help of many field technicians who collected data. Dr. Jeff

Mosley and Dr. Jay Rotella, and Montana Fish, Wildlife and Parks (MT FWP) staff John Ensign and Melissa Foster, provided useful guidance throughout the study. Associate Editor Dr. Walter, Dr. Joseph Smith, and an anonymous reviewer provided comments that improved the manuscript.

## Author Contributions

**Conceptualization:** Megan C. Milligan, Lorelle I. Berkeley, Lance B. McNew.

**Data curation:** Megan C. Milligan.

**Formal analysis:** Megan C. Milligan.

**Funding acquisition:** Lorelle I. Berkeley, Lance B. McNew.

**Methodology:** Megan C. Milligan.

**Project administration:** Megan C. Milligan, Lance B. McNew.

**Writing – original draft:** Megan C. Milligan.

**Writing – review & editing:** Lorelle I. Berkeley, Lance B. McNew.

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
