## [Decision Letter · Decision Letter 0]

13 Feb 2020

PONE-D-19-33753

Habitat selection of female sharp-tailed grouse in grasslands managed for livestock production

PLOS ONE

Dear Dr. Milligan,

Thank you for submitting your manuscript to PLOS ONE. After careful consideration, we feel that it has merit but does not fully meet PLOS ONE’s publication criteria as it currently stands. Therefore, we invite you to submit a revised version of the manuscript that addresses the points raised during the review process.

We would appreciate receiving your revised manuscript by Mar 29 2020 11:59PM. To enhance the reproducibility of your results, we recommend that if applicable you deposit your laboratory protocols in protocols.io, where a protocol can be assigned its own identifier (DOI) such that it can be cited independently in the future. For instructions see: http://journals.plos.org/plosone/s/submission-guidelines#loc-laboratory-protocols

We look forward to receiving your revised manuscript.

Kind regards,

W. David Walter, Ph.D.

Academic Editor

PLOS ONE

Additional Editor Comments (if provided):

In addition to suggestions provided by the reviewers, please pay particular attention to those comments on the details of your second-order selection. Please present those details requested, and, in your potential resubmission, please address why 2 different methods were used for second-order selection (compositional analysis) and third-order selection (RSF) because they present 2 very different attempts at assessing habitat selection.

Journal Requirements:

2. Our internal editors have looked over your manuscript and determined that it is within the scope of our Biodiversity Conservation Call for Papers. This collection of papers is headed by a team of Guest Editors for PLOS ONE (https://collections.plos.org/s/biodiversity). The Collection will encompass a diverse range of research articles on biodiversity conservation, including conservation of endangered species. Additional information can be found on our announcement page: https://collections.plos.org/s/biodiversity

If you would like your manuscript to be considered for this collection, please let us know in your cover letter and we will ensure that your paper is treated as if you were responding to this call. If you would prefer to remove your manuscript from collection consideration, please specify this in the cover letter.

3. In your Methods section, please provide additional location information of the study area, including geographic coordinates for the data set if available.

Reviewers' comments:

Reviewer's Responses to Questions

**Comments to the Author**

1. Is the manuscript technically sound, and do the data support the conclusions?

Reviewer #1: Partly

Reviewer #2: Yes

2. Has the statistical analysis been performed appropriately and rigorously? 

Reviewer #1: No

Reviewer #2: I Don't Know

3. Have the authors made all data underlying the findings in their manuscript fully available?

Reviewer #1: Yes

Reviewer #2: Yes

4. Is the manuscript presented in an intelligible fashion and written in standard English?

Reviewer #1: Yes

Reviewer #2: Yes

5. Review Comments to the Author

Reviewer #1: Review of “Habitat selection of female” by Milligan et al.

This is well-written manuscript on a topic and species of considerable management interest. Specific comments follow, referenced by line number.

Line 14: Habitat selection affects much more than abundance, as the authors point out in the introduction. It might be helpful to add or change a word in this line to acknowledge that (something along the lines of performance, viability, or persistence).

Line 16: Consider “…thus influencing habitat selection and population size” (or performance, viability, or persistence).

Line 20: Mention where the research took place.

Line 29: I suggest deleting the word “moderate,” simply stating that we want stocking rates that maintain native rangelands.

Line 65: Here and throughout the paper, many of the cited references are about greater prairie-chickens, not sharp-tailed grouse. I realized they are congenerics and that more information is available about chickens than sharptails, but I feel like there is a bit of a bait-and-switch by mentioning sharptails in some instances and grouse (accompanied by chicken papers) in others. Many solid papers about sharptail habitat selection and reproduction are not cited in this manuscript. In addition, management, environment, and social factors differ considerably between the authors’ study area and the Flint Hills, and the paper would be improved if more background was provided for the study area.

Line 107: Clarification on some of the metrics would be good. The paper mentions “unfragmented grassland patches,” but that’s a bit of an oxymoron to me. Landscape ecologists think of habitat patches that, depending on patch size and arrangement, constitute a fragmented or unfragmented landscape. If patchiness is determined by grazing regime, then it would be more clear to refer to heterogeneity or something similar.

Line 199: A major shortcoming to compositional analysis is that it only considers composition of the landscape and doesn’t consider things like proximity to roads, wells, edge, etc. I think an RSF-based approach (similar to the next section but with a different set of availabilities) would be much more informative.

Line 214: How the available points were determined should be described, as their number and spatial distribution can substantially affect results.

Line 220: Perhaps “scale” instead of “spatial grain,” which is frequently used to describe the resolution of landcover data.

Line 222: Good!

Line 234: I’m confused, as line 210 indicates that Design II of Manly et al. was used. Manly et al. states that “With design III studies the use and availability of resource units is measured separately for each animal.” Random effects as espoused by reference 62 (Gillies et al.) are used when “individual animals are monitored and pooled to estimate population level effects.” I’ll readily admit that I’m easily confused, but I think some clarification as to what is being assessed and how it was done is necessary.

Lines 279-280: Table 2 also confuses me, as it provides cumulative wi, but all the models (other than the null) only have one variable as evidenced by a k of 3. I suggest deleting the column with the cumulative wi or explaining the purpose of that column.

Lines 289-290: The “Total” row should be eliminated. Simply state that there was a total of 142 females. The other values appear to be means, except for Max area, which is not the mean but a copy of the value for rest-rotation.

Lines 292-293: The compositional analysis doesn’t really provide a whole lot of information that would be helpful for management.

Line 302: As mentioned before, “grain size” should be replaced. Here it suggests an area of 1300m x 1300m, but in reality, 1300 m is the radius of a circle with an area 3.14 times larger than the implied square.

Line 315: What about the signs of the relationships in Figure 3?

Line 319: Figure 4 indicates that females select for proximity to roads, which is quite different from the abstract, which states that females select for large intact grasslands.

Lines 348-350: Good point.

Lines 388-401: This paper focuses on habitat selection, but it would be useful if the authors provided more information from other aspects of their research regarding how reproductive success was influenced by habitat, and if the same factors popped out in both analyses.

Reviewer #2: Overall, this is a nicely set-up study that seeks to answer some pertinent, long-unanswered questions about effects of livestock management on a sharp-tailed grouse and their breeding habitat. The paper is generally well written and the conclusions appear well-supported by the results. I do, however, have a couple substantial issues with the analysis and its interpretation. These concerns are related and may be cleared up fairly quickly.

First, the third-order resource selection model seems to be misspecified. Random effects seem to have been included in a way that makes no biological or mathematical sense.

Second, the considerable space is given to interpreting these random effects as ‘individual variation’ in behavior, which is an interpretation that the model structure—if I’m seeing it right—doesn’t support.

I’m having a hard time knowing what I’m looking at in the results because the methods lack important details about the model. I would like to see the methods clarified by including full model statements (including priors—the reader is told that the priors are vague or uninformative, but has no way of verifying that for themselves because they are not provided) and, ideally, JAGS model code in Supplemental Materials.

If I am correct in my understanding that random intercepts, but not random slopes, were included in this RSF model, then I would like to see this corrected by either a) removing the random intercepts and any references to variation among individuals in their behavior, since this wasn’t a feature of the model, or b) adding random slopes and interpreting those parameters as individual variation.

Option A would take hardly any time at all, whereas Option B would entail a bit of work. I’m not set on either one, but I do tend to think a random-intercept/random-slope model is more appropriate for this study design.

Best regards,

Joe Smith

Line-by-line comments are provided as an attachment.

6. PLOS authors have the option to publish the peer review history of their article (what does this mean?). If published, this will include your full peer review and any attached files.

Reviewer #1: No

Reviewer #2: Yes: Joseph Smith

---

## [Author Response · Author response to Decision Letter 0]

19 Mar 2020

Reviewer #1: Review of “Habitat selection of female” by Milligan et al.

This is well-written manuscript on a topic and species of considerable management interest. Specific comments follow, referenced by line number.

Line 14: Habitat selection affects much more than abundance, as the authors point out in the introduction. It might be helpful to add or change a word in this line to acknowledge that (something along the lines of performance, viability, or persistence).

Author response: We agree and have added additional wording to convey that habitat selection affects more than just abundance (line 18).

Line 16: Consider “…thus influencing habitat selection and population size” (or performance, viability, or persistence).

Author response: We have changed the wording as suggested (lines 20-21).

Line 20: Mention where the research took place.

Author response: We have added the location of our study area (lines 25-26).

Line 29: I suggest deleting the word “moderate,” simply stating that we want stocking rates that maintain native rangelands.

Author response: We feel that it is important to include the word moderate to avoid overstating our conclusions, particularly given the number of studies that have found negative effects of high stocking rates (Briske et al. 2008, McNew et al. 2015, Kraft 2016, Winder et al. 2018). Stocking rates throughout our study area were generally moderate and it is possible that wildlife responses may be different at higher stocking rates. Therefore, we have kept the wording to qualify our conclusions.

Line 65: Here and throughout the paper, many of the cited references are about greater prairie-chickens, not sharp-tailed grouse. I realized they are congenerics and that more information is available about chickens than sharptails, but I feel like there is a bit of a bait-and-switch by mentioning sharptails in some instances and grouse (accompanied by chicken papers) in others. Many solid papers about sharptail habitat selection and reproduction are not cited in this manuscript. In addition, management, environment, and social factors differ considerably between the authors’ study area and the Flint Hills, and the paper would be improved if more background was provided for the study area.

Author response: We agree that it would be better to have more citations from sharp-tailed grouse papers and we have tried to include them wherever possible (e.g., Christenson 1970, Hillman and Jackson 1973, Marks and Marks 1987, Cope 1992, Saab and Marks 1992, Roersma 2001, Manzer 2004, Boisvert et al. 2005, Goddard et al. 2009, Stonehouse et al. 2015). However, there is no previous research on sharp-tailed grouse habitat selection in relation to grazing management, which is why we had to rely on the prairie-chicken literature, particularly studies in the Flint Hills, which have evaluated the relationship between grouse and grazing in depth.

Line 107: Clarification on some of the metrics would be good. The paper mentions “unfragmented grassland patches,” but that’s a bit of an oxymoron to me. Landscape ecologists think of habitat patches that, depending on patch size and arrangement, constitute a fragmented or unfragmented landscape. If patchiness is determined by grazing regime, then it would be more clear to refer to heterogeneity or something similar.

Author response: We agree that our use of the term “fragmented” is confusing because it is scale-dependent and grasslands are naturally heterogeneous. In this context, we were referring to grassland fragmentation caused by other land uses, such as cropland. We have clarified that we hypothesized that grouse would select for large grassland patches and removed reference to “unfragmented patches” (line 111).

Line 199: A major shortcoming to compositional analysis is that it only considers composition of the landscape and doesn’t consider things like proximity to roads, wells, edge, etc. I think an RSF-based approach (similar to the next section but with a different set of availabilities) would be much more informative.

Author response: We agree that one major limitation of compositional analysis is that it cannot accommodate continuous predictors. However, compositional analysis has several advantages, particularly that it assesses proportional habitat use and so avoids the unit-sum problem and that it correctly uses the individual home range as the sample unit. Although it was originally proposed in 1993, compositional analysis is still widely used in published articles today (over 50 articles citing the original paper on Google Scholar since 2019, for example). In addition, it was an appropriate approach given our objectives to evaluate home range selection relative to landcover type and grazing system, both of which are categorical variables. We have clarified both the approach and its usefulness given our objectives in the methods section (lines 214-219).

Line 214: How the available points were determined should be described, as their number and spatial distribution can substantially affect results.

Author response: We have clarified that available points were sampled randomly through an individual’s home range (lines 222-224).

Line 220: Perhaps “scale” instead of “spatial grain,” which is frequently used to describe the resolution of landcover data.

Author response: We have changed the wording throughout to use “scale” instead of “spatial grain”.

Line 222: Good!

Line 234: I’m confused, as line 210 indicates that Design II of Manly et al. was used. Manly et al. states that “With design III studies the use and availability of resource units is measured separately for each animal.” Random effects as espoused by reference 62 (Gillies et al.) are used when “individual animals are monitored and pooled to estimate population level effects.” I’ll readily admit that I’m easily confused, but I think some clarification as to what is being assessed and how it was done is necessary.

Author response: We stated that we used Design III of Manly et al (2002), where the “animals in a sample are radio-collared, and the relocations of an animal identify used resource units for that animal. . . The collection of available or unused resource units within an animal’s home range is sampled or censused.” We have included additional language to clarify that individual telemetry locations were used locations and we randomly sampled available locations within each individual’s home range (lines 220-225). 

Lines 279-280: Table 2 also confuses me, as it provides cumulative wi, but all the models (other than the null) only have one variable as evidenced by a k of 3. I suggest deleting the column with the cumulative wi or explaining the purpose of that column.

Author response: This is a good point and we have removed the column showing cumulative weights. 

Lines 289-290: The “Total” row should be eliminated. Simply state that there was a total of 142 females. The other values appear to be means, except for Max area, which is not the mean but a copy of the value for rest-rotation.

Author response: We have removed the “Total” row.

Lines 292-293: The compositional analysis doesn’t really provide a whole lot of information that would be helpful for management.

Author response: Given the strong selection for mixed-grass prairie, we feel that the compositional analysis provides valuable evidence for the importance of large grassland patches. We have clarified the language to stress that we mean large patches and not unfragmented patches, given that the term “fragmented” could mean multiple things in this context. This should clarify that our results suggest that conserving large grassland patches from conversion to other land uses is an appropriate and useful management strategy for the conservation of sharp-tailed grouse.

Line 302: As mentioned before, “grain size” should be replaced. Here it suggests an area of 1300m x 1300m, but in reality, 1300 m is the radius of a circle with an area 3.14 times larger than the implied square.

Author response: We have changed the wording throughout to use “scale” instead of “spatial grain”.

Line 315: What about the signs of the relationships in Figure 3?

Author response: We have redone our third order analysis and now included a graph showing the mean and 95% credible intervals for each population-level selection coefficient in the analysis which shows the direction and strength of the relationships (Fig 3).

Line 319: Figure 4 indicates that females select for proximity to roads, which is quite different from the abstract, which states that females select for large intact grasslands.

Author response: Although this relationship was very weak, it was no longer significant after we reanalyzed the data following the second reviewer’s comments. 

Lines 348-350: Good point.

Lines 388-401: This paper focuses on habitat selection, but it would be useful if the authors provided more information from other aspects of their research regarding how reproductive success was influenced by habitat, and if the same factors popped out in both analyses.

Author response: We have included references to other aspects of our research evaluating both adult survival and reproduction to place our results in context and inform the reader how other aspects of grouse ecology were influenced by the same or different habitat variables (lines 342-344, lines 349-350, lines 371-372).

Reviewer #2: Overall, this is a nicely set-up study that seeks to answer some pertinent, long-unanswered questions about effects of livestock management on a sharp-tailed grouse and their breeding habitat. The paper is generally well written and the conclusions appear well-supported by the results. I do, however, have a couple substantial issues with the analysis and its interpretation. These concerns are related and may be cleared up fairly quickly.

First, the third-order resource selection model seems to be misspecified. Random effects seem to have been included in a way that makes no biological or mathematical sense.

Second, the considerable space is given to interpreting these random effects as ‘individual variation’ in behavior, which is an interpretation that the model structure—if I’m seeing it right—doesn’t support.

I’m having a hard time knowing what I’m looking at in the results because the methods lack important details about the model. I would like to see the methods clarified by including full model statements (including priors—the reader is told that the priors are vague or uninformative, but has no way of verifying that for themselves because they are not provided) and, ideally, JAGS model code in Supplemental Materials.

If I am correct in my understanding that random intercepts, but not random slopes, were included in this RSF model, then I would like to see this corrected by either a) removing the random intercepts and any references to variation among individuals in their behavior, since this wasn’t a feature of the model, or b) adding random slopes and interpreting those parameters as individual variation.

Option A would take hardly any time at all, whereas Option B would entail a bit of work. I’m not set on either one, but I do tend to think a random-intercept/random-slope model is more appropriate for this study design.

Author response: Thank you for the excellent review and recommendations. We have followed Option B and redone our analysis with random slopes, which allows us to interpret those parameters as individual variation. We have also included model code in the Supplemental Materials so that a reader can fully understand how the model was specified.

Line-by-line comments:

Line 25: May need to remove reference to individual-level variation in habitat selection; see

detailed comments on methods below.

Author response: Given that we have now included random slopes in our model, we can make inferences regarding individual-level variation in habitat selection.

Line 39: I’m being nitpicky here, but this is such a vague, fluffy statement that it could mean

almost anything. Perhaps a specific example of how ‘conservation and management actions’

could ‘consider’ this information would imbue this statement with some needed substance.

Author response: We have removed this sentence.

Line 54: Make the speculative/hypothetical nature of this claim perfectly clear here. The two

provided citations provide no actual evidence that they function like patch-burn grazing. The

UGBEP makes a pretty unequivocal statement that they benefit upland birds, but provides no

evidence backing that up whatsoever. Frisina’s paper only deals with whether rotational grazing

can reduce conflict between cattle and elk on winter range, and never mentions heterogeneity.

Just make sure you don’t accidentally help propagate misinformation by leading the reader

(most of whom don’t check citations) to believe that there are studies that support this.

Line 61: Same comment here as above. Yes, they will have increased residual cover, but I don’t

think I’ve ever seen any evidence that RGS increase structural heterogeneity or create a

‘patchwork’ relative to other grazing systems (I haven’t searched exhaustively, though). We

actually found the opposite effect in Roundup, and it looks like heterogeneity in VOR was lowest

on RGS ranches in your study, too (Fig 6, random plots, in your recent REM paper--you could

even cite that). You could simply add a statement like, “but this hypothesis has not been

adequately tested.”

I think the purpose of RGS may have been lost in translation between range managers and

wildlifers. In Bailey and Brown (2011) Rangeland Ecol Manage 64:1–9, they write, “To alleviate

selective grazing across multiple scales, many rangeland managers have implemented

rotational grazing systems (RGS)...The potential benefits are projected to derive from a

decrease in area available to livestock at any given time (improved distribution); an increase in

stock density (increased uniformity of defoliation across species and communities)...”

So, rangeland managers may be quite aware that these systems are having the exact opposite

effect, at least in some settings.

Author response: We have changed the wording as suggested to emphasize that this is a hypothetical relationship (line 58).

Lines 70–71: This sentence is also very vague. What kind of ‘implications?’

Author response: We have changed the wording to be specific about the kind of implications (lines 73-74). 

Line 185: Define an ‘abrupt change.’

Author response: We have changed the wording to more clearly define how we measured edge density (lines 189-191).

Line 200: I’d like to see more detail on your methods for the second-order resource selection

analysis, rather than just referring the reader to a couple citations. Which variables were

considered, how was model fitting and model selection accomplished? There’s no way I could

re-create your analysis with the details you give here.

Author response: We have included additional details on both variables and models considered in the analyses assessing home range size and second order selection. We included additional information on which models were evaluated and our model selection procedure for the home range size analysis and we have included further details on compositional analysis to make the process more transparent (lines 181-219).

Lines 232–234: “For all models, we used generalized linear mixed models in a Bayesian

framework with a logit-link and female ID as a random intercept to account for potential

autocorrelation among sampling points [62, 63].”

I’m pretty sure this isn’t doing what you intended. Given your particular availability sampling design, these random intercepts are unnecessary and shouldn’t affect your inference at all. You later (lines 253–255) describe σ 2 as representing individual-level variation in selection for various habitat variables, which is a bit different than the way you justify inclusion of random

intercepts here (autocorrelation in the response variable--not sure what that means in this case),

but this is not what random intercepts do, either.

Consider what the intercept represents in a simple, univariate logit-link resource selection

function where you’ve centered your covariate at zero. If you convert it to the probability scale,

it’s the predicted probability that a case/sample is a used case (1) rather than an available case

(0) at the mean value of the covariate. In a use-availability RSF, its value is going to reflect the

prevalence of use cases—your ratio of used to available samples. You determined that when

you decided how many random samples to include for each used sample. The intercept has no

biological meaning in an RSF, it is simply a reflection of the way the investigator sets up the

analysis.

Unless you included both random intercepts and random adjustments to the slope (coefficient for the covariate—e.g., ,where the j subscript indexes individuals, βj ~ N(β, σ2) from your citation (#69), then all you are doing by allowing the intercept to vary among individuals is allowing random adjustments to the overall (i.e., regardless of the covariate value) probability of use.

Given this is a design parameter, not a biological one, and in your study design you’ve fixed it at 1:15 = 0.0625 for all individuals in your dataset (this is why your y-axis tops out at about 0.07 in Fig. 4), there is no reason to expect that there will be any variation among individuals at all. By design, individuals do not vary in mean P(use) (i.e., prevalance), and you’ve fixed the regression coefficients, so they don’t vary in their responses to covariates. So what is the variation you’re modeling with these random effects? You can prove this to yourself by running this R code with your data:

require(here)

require(tidyr)

require(dplyr)

require(lme4)

df <- read.csv(here('data/Pts.15.FULL.csv'), stringsAsFactors = FALSE)

# check that prevalence is equal across individuals

df %>%

select(id, use) %>%

group_by(id) %>%

summarize(prevalence = mean(use)) %>%

pull(prevalence)

# scale and center continuous covariates

df.scaled <- df %>% mutate_if(is.double, scale)

# fit a simple univariate fixed-effects glm

m.fixed <- glm(use ~ 1 + roaddist, data=df.scaled, family=binomial(link = ‘logit’))

# fit the same model, but with random intercepts for individuals

m.rand.intercept <- glmer(use ~ 1 + roaddist + (1|id), data=df.scaled,

family=binomial(link = ‘logit’))

# note warning about boundary effect

# check estimates and SE's of fixed effects (your inference)

summary(m.fixed)

summary(m.rand.intercept)

The SEs and coefficient estimates are equal for these two model specifications. Your inference is exactly the same. Also look at the estimate of individual-level variation for the glmer--it’s effectively zero (the boundary effect).

The random intercept takes on a very different role when you also include random slopes. If individuals vary in their selection for a covariate, then you need random intercepts because it doesn’t make sense to force P(use) through a single point along the x-axis. If some respond positively and others negatively to distance to roads, for example, there’s no reason to expect they’ll all converge on the exact same probability of selection at the mean distance to roads in your dataset. Random intercepts give some needed flexibility there.

Your random intercepts, however, have no clear biological interpretation that I can think of, which is probably why your results show they’re all about the same, may in fact be zero, and, if you plotted their densities instead of means and CRIs, you’d probably find them to closely reflect their priors (which I’m guessing were Uniform(0,2) based on Fig. 5). This is a really long-winded way of saying you need random slopes, not just random intercepts, if you want to know anything about individual-level variation in selection. If you don’t, you should take the random intercepts out because prevalence is a design parameter that you’ve fixed to be constant across individuals (Line 231). They are not affecting your inference.

Author response: While random intercepts were necessary given our unbalanced sampling design (individuals had different numbers of relocations), we have redone this analysis to now incorporate random slopes in addition to random intercepts. 

Line 243: “Regression coefficients for each variable were the product of binary indicator

variables and both continuous and categorical covariates.”

Hmmm, this doesn’t make sense the way it’s worded. They should be the product of the

indicator variable and an effect size parameter (the ‘raw’ coefficient). Multiplying the indicator by the covariate (the whole vector? I’m having a hard time picturing what you mean here) will give you either a vector of zeros or your original vector of covariates. It won’t give you a regression coefficient.

Author response: Now that we have redone this analysis, we have removed this statement.

Line 246: “We assumed that all variables with high inclusion probability based on the posterior

distributions of their indicator variables influenced habitat selection and variables with inclusion

probabilities ≤ 0.25 were unimportant [67]” 

Mutshinda et al. used ≤0.25 to identify unimportant variables because they used a fixed prior probability of 0.5, so that posterior probability coincides with a Bayes factor of 0.33, or 3:1 against the variable. Because you have a distribution of prior odds, rather than a fixed value, 0.25 doesn’t have any real meaning here. If, for example, your model converged on an inclusion probability of 0.1, then 0.25 would actually represent a Bayes factor of 3, or 3:1 evidence for the variable. If anything, I could see taking the median posterior inclusion probability and using that to calculate the cutoff, but that seems a little circular, too. I’d argue that variable selection is simpler if you just pick what you think is an appropriate prior probability and consider it fixed. With the amount of data you have, I doubt this value is going to have much of an influence on which variables are identified as important, but you could always try running it with a few values to check that sensitivity. Unless mean variable inclusion probability is a parameter you’re interested in making inference on (which it doesn’t seem like it is, given that you don’t report it in the results), then I don’t quite see the value in putting a prior on it and estimating it.

Author response: We have redone the analysis and are no longer using indicator variables.

Line 250: Define “standardized coefficients.”

Author response: We have clarified that we centered and scaled all predictor variables to calculate standardized coefficients (lines 255-257).

Line 252: So are you using credible intervals or the posterior inclusion probability to determine

which variables are important? Is one given higher priority than another?

Author response: We are no longer using inclusion probabilities and have clarified that we used 95% credible intervals to determine which variables were important (lines 257-258).

Line 253: I’m not exactly sure what these σ2 are, but given that you did not include random

slopes, they cannot represent individual variation in behavior.

Author response: We have now included random slopes and make inferences on individual variation using the variation in individual-specific slopes for each variable (Fig 4) and the number of individuals that were selecting for and against each variable (Fig 5).

Line 259: “Vague uniform or normal priors were used for all model parameters related to

covariates and their measures of error.”

This statement is itself a little vague. I would like to see either a) complete model statements in your methods section or b) your JAGS model code (in supplemental info?) so I can see exactly what priors were placed on each parameter. Ideally, provide both.

Author response: This is a good point and we have now included model code in our Supplemental Material so that the reader can fully understand our methods.

Line 269: Insert ‘Gelman-Rubin statistic’ before ‘values’ or just refer to them as ‘potential scale

reduction factors’ so we know what values you’re referring to.

Author response: Given our new analysis approach (fitting a model with integrated nested Laplace approximation), this statistic is no longer relevant as it applies to MCMC approaches.

Line 270: I’m glad to see you assessed model fit, but it would be nice to know which specific

‘attribute of the data’ you used to calculate the Bayesian p -value.

Author response: We have removed this statement given our new analysis approach.

Line 290: Were any models with >1 predictor considered? Again, more details on the 2nd order

resource selection analysis are needed.

Author response: We have included additional details on both variables and models considered in the analyses assessing home range size and second order selection. We included additional information clarifying that only single-variable models were evaluated and our model selection procedure for the home range size analysis and we have included further details on compositional analysis to make the process more transparent (lines 181-219).

Lines 292–293: This sentence and Table 3 seem to provide exactly the same information, so I

would just include one or the other. I find Table 3 hard to interpret (isn’t the info on either side of the diagonal redundant? And why isn’t there a #2-ranked landcover?), so I’d suggest axing it.

Author response: This is a good point and we have now removed Table 3.

Line 308: Again, I don’t think 0.25 is a meaningful cutoff. Try to come up with a criterion more

applicable to your particular model.

Author response: With our revised analysis, we are no longer using indicator variables and so no longer require a cutoff.

Line 311: This doesn’t indicate individual variation in habitat selection unless you fit random

intercept and random slope models. So am I right to assume these variances indicate variation

among individual-level adjustments to the intercept? If so, why is there more than one? Are

these from separate, univariate models? I’m generally very confused about your model

structure, so I’ll reiterate here that it would really help if you provided model statements and

code so readers know exactly what parameters these are. Also (and this is a personal preference thing unless the journal has specific guidelines about it), I’m much more accustomed to thinking of variation on the scale of standard deviations rather than variances. Seems easier to interpret since it’s on the same scale as the regression coefficients and data. Is there a specific reason you report σ 2 rather than σ?

Author response: Now that we have included random slopes in our model, we make inferences about individual variation using the variation in individual-specific slopes for each variable (Fig 4). We have also included model code in our Supplemental Material so readers know exactly how the model was specified. 

Line 377: Given that you have the ability to fit random-slope, random-intercept models (because

you sampled availability at the individual level), there is no reason you shouldn’t. In fact, I would argue that assuming the slopes are fixed across individuals is inappropriate given that

availability differed across your study area, as you state on Line 348. At the individual level and

the third-order, you are only likely to detect avoidance of, say, oil and gas features, if there are

oil and gas features within the home range to avoid. By forcing all the responses to be the same

among individuals, you’re not allowing your models to pick up on those potential functional

responses to availability. Moreover, fitting random-slope, random-intercept models does not preclude making inference at the population level. The beauty of these multi-level models is that you can make inference at multiple levels of the hierarchy—population or individual. Going back to the example of citation #69, this model provides not only estimates of the individual-level σ j ’s, but also the population-level β .

Author response: This is a good point and we have now included random slopes.

Fig. 1. Might this look a little better on the log(ha) scale?

Author response: We feel it is more intuitive if the variable is not transformed, but can defer to the associate editor.

Fig. 2. Since both variables are continuous, why not plot the data? It’s a more honest way of

showing the strength and predictive capability of a relationship than just plotting the regression

line for the mean and its confidence interval.

Author response: We have now included the raw data in our plot of the regression results (Fig 2).

Fig. 3 and Fig. 5: Both of these figures depict information that seems better suited to a table.

Just a suggestion.

Author response: We have redone our analysis, so Fig 3 and Fig 5 now present different information.

---

## [Decision Letter · Decision Letter 1]

22 Apr 2020

PONE-D-19-33753R1

Habitat selection of female sharp-tailed grouse in grasslands managed for livestock production

PLOS ONE

Dear Dr. Milligan:

Thank you for submitting your manuscript to PLOS ONE. After careful consideration, we feel that it has merit but does not fully meet PLOS ONE’s publication criteria as it currently stands. Therefore, we invite you to submit a revised version of the manuscript that addresses the points raised during the review process.

We would appreciate receiving your revised manuscript by 22 May 2020. To enhance the reproducibility of your results, we recommend that if applicable you deposit your laboratory protocols in protocols.io, where a protocol can be assigned its own identifier (DOI) such that it can be cited independently in the future. For instructions see: http://journals.plos.org/plosone/s/submission-guidelines#loc-laboratory-protocols

We look forward to receiving your revised manuscript.

Kind regards,

W. David Walter, Ph.D.

Academic Editor

PLOS ONE

Additional Editor Comments (if provided):

We thank the authors for addressing the reviewer comments in great detail. I would request the authors address the remaining concerns of the same 2 authors as well as my own concerns regarding second-order selection.

In the previous draft, a reviewer requested more details on second-order selection, particularly on the variables and reason for not conducting RSF for second-order as done with third-order. I don’t believe the authors adequately addressed the previous concerns but more importantly the results seem confusing to me. Let’s start with the justification and the response from the authors in the previous version:

Line 200: I’d like to see more detail on your methods for the second-order resource selection

analysis, rather than just referring the reader to a couple citations. Which variables were

considered, how was model fitting and model selection accomplished? There’s no way I could

re-create your analysis with the details you give here.

Author response: We have included additional details on both variables and models considered in the analyses assessing home range size and second order selection. We included additional information on which models were evaluated and our model selection procedure for the home range size analysis and we have included further details on compositional analysis to make the process more transparent (lines 181-219).

AE’s comments: The authors state on Lines 210: “We used compositional analysis to compare used versus available landcover types and grazing systems separately.” However, in the results the authors identify results only for landcover types (Lines 295-301) in the order of preference. They then claim in Lines 301-303 that “There was no evidence that selection of home ranges in relation to grazing system was different from random (p = 0.20), suggesting that females were not differentiating between pastures in the different grazing systems.” This appears to be relating the linear models in the preceding section linking size of home range to grazing system, not compositional analysis? If it were a separate analysis on grazying system, I would expect a similar sentence of the 3 systems as presented for the landcover types in Lines 295-301 (i.e., no preference for grazing system). I am not sure if I am missing something here?

More importantly, your justification for second-order selection that “others are still using it” is poor and I recommend the authors delete the entire concept of second order selection in your manuscript. Although the authors present compositional analysis in the Methods and Results, the authors then combine both second- and third-order selection in a statement in Lines 357-358 and second-order selection is never mentioned again. The details of second-order selection does not add much obvious value to the manuscript and the authors do not even spend any time on its contribution in the Discussion?

Reviewers' comments:

Reviewer's Responses to Questions

**Comments to the Author**

1. If the authors have adequately addressed your comments raised in a previous round of review and you feel that this manuscript is now acceptable for publication, you may indicate that here to bypass the “Comments to the Author” section, enter your conflict of interest statement in the “Confidential to Editor” section, and submit your "Accept" recommendation.

Reviewer #1: (No Response)

Reviewer #2: (No Response)

2. Is the manuscript technically sound, and do the data support the conclusions?

Reviewer #1: Yes

Reviewer #2: Partly

3. Has the statistical analysis been performed appropriately and rigorously? 

Reviewer #1: Yes

Reviewer #2: Yes

4. Have the authors made all data underlying the findings in their manuscript fully available?

Reviewer #1: Yes

Reviewer #2: Yes

5. Is the manuscript presented in an intelligible fashion and written in standard English?

Reviewer #1: Yes

Reviewer #2: Yes

6. Review Comments to the Author

Reviewer #1: Lines 32-33: Very minor point of subject-verb agreement: “…strategies…are important for the conservation of sharp-tailed grouse. “

Line 69: Similar to a comment in my initial review, I find this type of citation to be misleading. Sharp-tailed grouse are “Recognized as an indicator species for grassland ecosystems” by an unpublished master’s thesis from Kansas on lesser prairie chickens? Yes, sharptails are recognized as indicator/keystone/umbrella species, but there are certainly more appropriate entities to cite, including state wildlife management plans, joint venture implementation plans, and other plans developed by groups involved with on-the-ground conservation in the study area.

Lines 218-219: The objective might have been to evaluate landcover type and grazing system, but in many areas those are influenced by or correlated with continuous variables such as proximity to road, proximity to water, or topographic variation, to name a few. I still maintain that the compositional analysis leaves out important information. Variables quantifying anthropogenic disturbance are included in the within-home-range analysis (line 227); it only makes sense that if birds avoid such disturbance within their home range that they might avoid such disturbance when selecting a home range.

Line 289: Final decision is up to the editor, but most people find informative figure headings easier to understand than descriptive, i.e., the heading for Figure 2 would convey more information and be easier to assimilate if it read “Home range size was negatively related to density of edge habitat within the home range.” Other headings could be similarly revised.

Lines 349-352: But habitat selection across the landscape did not consider these factors, and the reason that some individuals had few to no roads or oil wells within their home range might be because they selected areas with few to no roads.

Lines 371-373: It was stated earlier that topography and soil type didn’t vary among grazing systems, but if there was substantial variation across the landscape (regardless of grazing system), might they have affected habitat selection?

Reviewer #2: Overall, I'm very happy with the improvements that were made in response to the first round of reviews. I'm especially glad the authors chose to re-analyze the third-order resource selection data with a varying-intercept and varying-slope model. There are just a few minor interpretation issues that I think should be addressed. These are discussed below.

Line-by-line comments on PONE-D-19-33753R1:

Line 190: This is a little too vague to reproduce. Is it edge length divided by home range size?

Line 209: You might consider using a term other than ‘home range’ in this context, since it doesn’t really apply to a group of individuals. Maybe ‘95% utilization distribution’ or something like that.

Line 275: I didn’t catch this before, but you must have had several females that contributed multiple home ranges to this analysis. In keeping with the rest of your analysis, you should account for this non-independence with random effects or subsample so each individual contributes just one home range.

Figure 1: Point taken re: interpretability, and I’ll defer to the AE’s judgement on this, too, but it’s awfully hard to see any differences among the means on this scale.

Figure 2: Need units on the x-axis label (m/km2, or whatever these are).

Line 308: Strike ‘the variable of’.

Figure 3: The y-axis label could be a lot more informative than ‘effect size.’ Maybe, ‘population level selection coefficient,’ or ‘global mean selection coefficient.’

Figure 4: Comparing this figure ot Figure 3, it’s apparent that these are not the individual-level slopes per se, but the individual-level adjustments to the population-level means shown in Figure 3. This should either be explained more clearly in the figure caption, or (and I’d prefer this) you could show boxplots of the actual individual-level slope estimates. This would show the location and scale of these slope parameters in one place.

Line 313 and Figure 5: This figure, and Figure 4, are interesting in that they show there is substantial individual-level variation in selection coefficients. But they immediately raise the question: Are we actually seeing individuals that vary widely in their habitat preferences, or are we just seeing more-or-less constant 3rd-order habitat use among home ranges with highly variable resource availability (i.e., functional responses, sensu Mysterud and Ims)? You could answer that with your data, but you kind of leave us hanging. And the statement on Line 313 that you’re seeing ‘no population-level selection’ kind of sweeps this issue under the rug.

Line 329: What you’ve shown here hasn’t convince me we are seeing ‘highly plastic habitat use.’ Keep in mind the distinction between resource use and resource selection. You’ve demonstrated highly variable selection coefficients, but you have not shown that individuals are highly variable in what they’re actually using. This comes back to my comment above re: functional responses to availability. Because you don’t test for functional responses, you haven’t convinced me these birds are actually displaying divergent behaviors/preferences when it comes to third order resource selection.

I’m not saying you’re interpreting your data wrong, just that you haven’t quite provided us, the readers, with enough evidence. I went ahead and plotted mean use (y-axis) against mean availability (x-axis) for several of your variables among home ranges, and this is what it looks like:

[see pdf version of comments!]

This tells me you’re probably right, they are very plastic in what they use. In fact, use seems to be more variable than availability in some cases. I’d suggest including something like this, or (even better) selection coefficients plotted against mean availability, if you want to make your argument of highly plastic habitat preference/use stronger. You could even combine the information in Figure 5 with this type of plot by color coding the home ranges according to positive, null, or negative individual-level selection coefficients.

Line 351: This sentence seems like a holdover from the last version of the manuscript, before you incorporated individual-level random slopes. Given what the plot above shows, I don’t think variation in availability among home ranges is obscuring some real response to oil pad distance—they just don’t seem to be avoiding them at all.

Line 361: Reconsider the word ‘inconsistent’ here; there’s nothing inconsistent about finding different habitat relationships for different species in a different ecological contexts.

Line 376: More important than what? I’m not picking up on your meaning here.

Line 380: This sentence also reads like a holdover from your last version.

Line 386: Clarify that you mean individual differences in resource selection. As I read your suggestion for future research here, it seems even more critical that you show readers that these differences in selection coefficients are not simply functional responses to availability. If they are, then your suggestion that they represent alternative strategies is much weaker.

Line 401: It’s up to you how much you want to synthesize the results of your various analyses for the readers of this paper, but I can’t help but think you’re leaving out some critical knowledge from your last paper in this conclusion. Just reading this, I’m left with the impression that grazing systems are irrelevant to sharp-tailed grouse in this ecosystem. But what about the fact that you showed nest success was significantly lower in these rotational grazing systems? If they’re not avoiding them, but they’re achieving lower reproductive success in them, then RGS aren’t just neutral or not quite as good as we’d hoped, they’re a (potentially) bad management practice for sharp-tailed grouse!

Again, this is largely a matter of personal preference, but I would personally love to see some synthesis of what you’re finding out there. Don’t assume people will search out all your research and put the puzzle together themselves.

Table S2: This table title should probably include more detail so it can stand alone. Something like: “Support for models predicting the spatial grain of each landcover variable that best predicts third-order female sharp-tailed grouse habitat selection during the breeding season, 2016–2018, based on deviance information criteria (DIC) values.

Table S3: Same comment as above.

7. PLOS authors have the option to publish the peer review history of their article (what does this mean?). If published, this will include your full peer review and any attached files.

Reviewer #1: No

Reviewer #2: Yes: Joseph Smith

---

## [Author Response · Author response to Decision Letter 1]

5 May 2020

Additional Editor Comments (if provided):

We thank the authors for addressing the reviewer comments in great detail. I would request the authors address the remaining concerns of the same 2 authors as well as my own concerns regarding second-order selection.

In the previous draft, a reviewer requested more details on second-order selection, particularly on the variables and reason for not conducting RSF for second-order as done with third-order. I don’t believe the authors adequately addressed the previous concerns but more importantly the results seem confusing to me. Let’s start with the justification and the response from the authors in the previous version:

Line 200: I’d like to see more detail on your methods for the second-order resource selection

analysis, rather than just referring the reader to a couple citations. Which variables were

considered, how was model fitting and model selection accomplished? There’s no way I could

re-create your analysis with the details you give here.

Author response: We have included additional details on both variables and models considered in the analyses assessing home range size and second order selection. We included additional information on which models were evaluated and our model selection procedure for the home range size analysis and we have included further details on compositional analysis to make the process more transparent (lines 181-219).

AE’s comments: The authors state on Lines 210: “We used compositional analysis to compare used versus available landcover types and grazing systems separately.” However, in the results the authors identify results only for landcover types (Lines 295-301) in the order of preference. They then claim in Lines 301-303 that “There was no evidence that selection of home ranges in relation to grazing system was different from random (p = 0.20), suggesting that females were not differentiating between pastures in the different grazing systems.” This appears to be relating the linear models in the preceding section linking size of home range to grazing system, not compositional analysis? If it were a separate analysis on grazying system, I would expect a similar sentence of the 3 systems as presented for the landcover types in Lines 295-301 (i.e., no preference for grazing system). I am not sure if I am missing something here?

More importantly, your justification for second-order selection that “others are still using it” is poor and I recommend the authors delete the entire concept of second order selection in your manuscript. Although the authors present compositional analysis in the Methods and Results, the authors then combine both second- and third-order selection in a statement in Lines 357-358 and second-order selection is never mentioned again. The details of second-order selection does not add much obvious value to the manuscript and the authors do not even spend any time on its contribution in the Discussion?

Author response: Thank you for these comments. We believe that it is important to include both second and third order selection because these analyses evaluate habitat selection at different spatial scales, but we have followed the first reviewer’s recommendation and changed our second order analysis so that it uses resource selection functions similar to the third order analysis rather than compositional analysis. This has allowed us to include additional continuous variables, such as distance to oil pad, which could not be accommodated with our previous analysis approach. We have outlined our approach in detail and believe that it adds important information to the manuscript.

Reviewer #1 

Lines 32-33: Very minor point of subject-verb agreement: “…strategies…are important for the conservation of sharp-tailed grouse. “

Author response: We have corrected this sentence (Line 33).

Line 69: Similar to a comment in my initial review, I find this type of citation to be misleading. Sharp-tailed grouse are “Recognized as an indicator species for grassland ecosystems” by an unpublished master’s thesis from Kansas on lesser prairie chickens? Yes, sharptails are recognized as indicator/keystone/umbrella species, but there are certainly more appropriate entities to cite, including state wildlife management plans, joint venture implementation plans, and other plans developed by groups involved with on-the-ground conservation in the study area.

Author response: This citation refers to a thesis on sharp-tailed grouse in Alberta that provided the best empirical evidence of which we are aware regarding sharp-tailed grouse as indicator species (Line 69). We agree that there are other plans that cite sharp-tailed grouse as indicator species, but they did not evaluate this concept, which is why we chose this citation. We can provide additional citations if the editor feels that is necessary.

Lines 218-219: The objective might have been to evaluate landcover type and grazing system, but in many areas those are influenced by or correlated with continuous variables such as proximity to road, proximity to water, or topographic variation, to name a few. I still maintain that the compositional analysis leaves out important information. Variables quantifying anthropogenic disturbance are included in the within-home-range analysis (line 227); it only makes sense that if birds avoid such disturbance within their home range that they might avoid such disturbance when selecting a home range.

Author response: Following this recommendation, we have redone our second order analysis and now use resource selection functions which evaluate all the variables included in the within-home range analysis.

Line 289: Final decision is up to the editor, but most people find informative figure headings easier to understand than descriptive, i.e., the heading for Figure 2 would convey more information and be easier to assimilate if it read “Home range size was negatively related to density of edge habitat within the home range.” Other headings could be similarly revised.

Author response: We can defer to the editor on this.

Lines 349-352: But habitat selection across the landscape did not consider these factors, and the reason that some individuals had few to no roads or oil wells within their home range might be because they selected areas with few to no roads.

Author response: Our second order analysis now includes these continuous variables, including roads and oil pads, so we have updated our discussion to reflect this.

Lines 371-373: It was stated earlier that topography and soil type didn’t vary among grazing systems, but if there was substantial variation across the landscape (regardless of grazing system), might they have affected habitat selection?

Author response: We have revised our analysis, so that it now includes additional variables which altered our discussion, so we have removed this sentence.

Reviewer #2: Overall, I'm very happy with the improvements that were made in response to the first round of reviews. I'm especially glad the authors chose to re-analyze the third-order resource selection data with a varying-intercept and varying-slope model. There are just a few minor interpretation issues that I think should be addressed. These are discussed below.

Line-by-line comments on PONE-D-19-33753R1:

Line 190: This is a little too vague to reproduce. Is it edge length divided by home range size?

Author response: We have clarified that edge density was edge length divided by home range size (Lines 196-198).

Line 209: You might consider using a term other than ‘home range’ in this context, since it doesn’t really apply to a group of individuals. Maybe ‘95% utilization distribution’ or something like that.

Author response: This is a good point and we clarified that the study area was defined by a utilization distribution, which is a more appropriate term than home range (Lines 205-206).

Line 275: I didn’t catch this before, but you must have had several females that contributed multiple home ranges to this analysis. In keeping with the rest of your analysis, you should account for this non-independence with random effects or subsample so each individual contributes just one home range.

Author response: We have updated our second order analysis and we only included one home range from each individual, which we clarified in the methods section (Lines 202-203).

Figure 1: Point taken re: interpretability, and I’ll defer to the AE’s judgement on this, too, but it’s awfully hard to see any differences among the means on this scale.

Author response: We can defer to the editor on this, but we found no evidence for a difference in home range size among grazing systems, which is what is depicted in this graph.

Figure 2: Need units on the x-axis label (m/km2, or whatever these are).

Author response: We have included units on the x-axis.

Line 308: Strike ‘the variable of’.

Author response: We have removed this phrase.

Figure 3: The y-axis label could be a lot more informative than ‘effect size.’ Maybe, ‘population level selection coefficient,’ or ‘global mean selection coefficient.’

Author response: We have revised the y-axis label following this recommendation.

Figure 4: Comparing this figure to Figure 3, it’s apparent that these are not the individual-level slopes per se, but the individual-level adjustments to the population-level means shown in Figure 3. This should either be explained more clearly in the figure caption, or (and I’d prefer this) you could show boxplots of the actual individual-level slope estimates. This would show the location and scale of these slope parameters in one place.

Author response: We have fixed this figure so that it now shows boxplots of the actual individual-level slope estimates.

Line 313 and Figure 5: This figure, and Figure 4, are interesting in that they show there is substantial individual-level variation in selection coefficients. But they immediately raise the question: Are we actually seeing individuals that vary widely in their habitat preferences, or are we just seeing more-or-less constant 3rd-order habitat use among home ranges with highly variable resource availability (i.e., functional responses, sensu Mysterud and Ims)? You could answer that with your data, but you kind of leave us hanging. And the statement on Line 313 that you’re seeing ‘no population-level selection’ kind of sweeps this issue under the rug.

Author response: This is a good point and we have included an additional figure (Figure 6) to allow readers to better evaluate our conclusions and to demonstrate that we are seeing highly flexible habitat selection, not a functional response.

Line 329: What you’ve shown here hasn’t convince me we are seeing ‘highly plastic habitat use.’ Keep in mind the distinction between resource use and resource selection. You’ve demonstrated highly variable selection coefficients, but you have not shown that individuals are highly variable in what they’re actually using. This comes back to my comment above re: functional responses to availability. Because you don’t test for functional responses, you haven’t convinced me these birds are actually displaying divergent behaviors/preferences when it comes to third order resource selection.

I’m not saying you’re interpreting your data wrong, just that you haven’t quite provided us, the readers, with enough evidence. I went ahead and plotted mean use (y-axis) against mean availability (x-axis) for several of your variables among home ranges, and this is what it looks like:

[see pdf version of comments!]

This tells me you’re probably right, they are very plastic in what they use. In fact, use seems to be more variable than availability in some cases. I’d suggest including something like this, or (even better) selection coefficients plotted against mean availability, if you want to make your argument of highly plastic habitat preference/use stronger. You could even combine the information in Figure 5 with this type of plot by color coding the home ranges according to positive, null, or negative individual-level selection coefficients.

Author response: Thank you for the useful suggestion. We have included a figure (Figure 6) like the one you suggested that plots resource use against availability and combined it with the previous Figure 5 to show the number of individuals selecting for and against specific habitat variables. This figure includes important information about the variation in individual selection and is similar to previously published figures evaluating functional responses.

Line 351: This sentence seems like a holdover from the last version of the manuscript, before you incorporated individual-level random slopes. Given what the plot above shows, I don’t think variation in availability among home ranges is obscuring some real response to oil pad distance—they just don’t seem to be avoiding them at all.

Author response: We have revised our discussion and removed this sentence.

Line 361: Reconsider the word ‘inconsistent’ here; there’s nothing inconsistent about finding different habitat relationships for different species in a different ecological contexts.

Author response: Given our updated results, we have revised this sentence (Line 368).

Line 376: More important than what? I’m not picking up on your meaning here.

Author response: We have revised our discussion and removed this sentence.

Line 380: This sentence also reads like a holdover from your last version. 

Author response: We have removed this sentence.

Line 386: Clarify that you mean individual differences in resource selection. As I read your suggestion for future research here, it seems even more critical that you show readers that these differences in selection coefficients are not simply functional responses to availability. If they are, then your suggestion that they represent alternative strategies is much weaker.

Author response: We have clarified that we mean individual differences in resource selection (Line 399).

Line 401: It’s up to you how much you want to synthesize the results of your various analyses for the readers of this paper, but I can’t help but think you’re leaving out some critical knowledge from your last paper in this conclusion. Just reading this, I’m left with the impression that grazing systems are irrelevant to sharp-tailed grouse in this ecosystem. But what about the fact that you showed nest success was significantly lower in these rotational grazing systems? If they’re not avoiding them, but they’re achieving lower reproductive success in them, then RGS aren’t just neutral or not quite as good as we’d hoped, they’re a (potentially) bad management practice for sharp-tailed grouse!

Again, this is largely a matter of personal preference, but I would personally love to see some synthesis of what you’re finding out there. Don’t assume people will search out all your research and put the puzzle together themselves.

Author response: This is a good suggestion and we have incorporated findings from previous research to better synthesize our results (Lines 370-373 and 404-406).

Table S2: This table title should probably include more detail so it can stand alone. Something like: “Support for models predicting the spatial grain of each landcover variable that best predicts third-order female sharp-tailed grouse habitat selection during the breeding season, 2016–2018, based on deviance information criteria (DIC) values.

Author response: We have altered the title to be more informative following this suggestion.

Table S3: Same comment as above.

Author response: We have altered the title to be more informative following this suggestion.

---

## [Editor Report · Decision Letter 2]

13 May 2020

Habitat selection of female sharp-tailed grouse in grasslands managed for livestock production

PONE-D-19-33753R2

Dear Dr. Milligan,

We are pleased to inform you that your manuscript has been judged scientifically suitable for publication and will be formally accepted for publication once it complies with all outstanding technical requirements.

With kind regards,

W. David Walter, Ph.D.

Academic Editor

PLOS ONE

Additional Editor Comments (optional):

I appreciate the authors edits to the previous reviews and their detailed responses to those reviews. Specifically, updating your compositional analysis to second-order selection with resource selection functions was much appreciated. I found a few items that I would appreciate the authors paying attention to in the type setting stage. As the authors are the experts on these topics, I just want to be sure they are presented properly in the final version of the manuscript.

Line 150: Should “lessees” be “leassees” or changed to “leaseholders” for clarity? Current spelling seems odd but if that is correct then please ignore my comment.

Lines 151: AUM? Spell out first use unless the authors believe this is known and understood worldwide?

Table 3: Remove the Cumulative AIC weight column as previously requested by a reviewer and I believe was confirmed removed by the authors in their rebuttal?

Line 309: Please change “Within the home range” to “At the third order” to be consistent and clear that this represents the start of the section on third-order selection of habitat.

Table S2 and S3: Spell out “Ag”to be clear it is agriculture. Also, Table S3 has S6 in the document?

Please be sure to go through all Tables throughout manuscript to check for inclusion and citing. Do the same for Supplemental Figures and Tables as well.
---

## [Editor Report · Acceptance letter]

21 May 2020

PONE-D-19-33753R2 

Habitat selection of female sharp-tailed grouse in grasslands managed for livestock production 

Dear Dr. Milligan:

I am pleased to inform you that your manuscript has been deemed suitable for publication in PLOS ONE. Congratulations! Your manuscript is now with our production department. 

With kind regards,

on behalf of

Dr. W. David Walter 

Academic Editor

PLOS ONE